# Actor-Critic Policy Optimization in a Large-Scale Imperfect-Information Game

Haobo Fu,[1][*] Weiming Liu,[2][*][†] Shuang Wu,[1] Yijia Wang,[3][†] Tao Yang,[1] Kai Li,[45] Junliang Xing,[45] Bin Li,[2] Bo Ma,[1] Qiang Fu,[1] and Wei Yang[1]

[1] Tencent AI Lab, Shenzhen, China
[2] University of Science and Technology of China, Hefei, China
[3] Peking University, Beijing, China
[4] Institute of Automation, Chinese Academy of Sciences, Beijing, China
[5] School of Artificial Intelligence, University of Chinese Academy of Sciences, Beijing, China

## Abstract

The deep policy gradient method has demonstrated promising results in many large-scale games, where the agent learns purely from its own experience. Yet, policy gradient methods with self-play suffer convergence problems to a Nash Equilibrium (NE) in multi-agent situations. Counterfactual regret minimization (CFR) has a convergence guarantee to a NE in 2-player zero-sum games, but it usually needs domain-specific abstractions to deal with large-scale games. Inheriting merits from both methods, in this paper we extend the actor-critic algorithm framework in deep reinforcement learning to tackle a large-scale 2-player zero-sum imperfect-information game, 1-on-1 Mahjong, whose information set size and game length are much larger than poker. The proposed algorithm, named Actor-Critic Hedge (ACH), modifies the policy optimization objective from originally maximizing the discounted returns to minimizing a type of weighted cumulative counterfactual regret. This modification is achieved by approximating the regret via a deep neural network and minimizing the regret via generating self-play policies using Hedge. ACH is theoretically justified as it is derived from a neural-based weighted CFR, for which we prove the convergence to a NE under certain conditions. Experimental results on the proposed 1-on-1 Mahjong benchmark and benchmarks from the literature demonstrate that ACH outperforms related state-of-the-art methods. Also, the agent obtained by ACH defeats a human champion in 1-on-1 Mahjong.

## 1 Introduction

Policy gradient methods using deep neural networks as policy and value approximators have been successfully applied to many large-scale games (Berner et al., 2019; Vinyals et al., 2019; Ye et al., 2020). Usually, a score function representing the discounted returns is maximized by the policy, i.e., the actor. In the meantime, a value function, known as the critic, is learned to guide the directions and magnitudes of the policy gradients. This type of actor-critic methods are efficiently scalable with regard to the game size and the amount of computational resources. However, as pointed out in Srinivasan et al. (2018) and Hennes et al. (2020), policy gradient methods with self-play have no convergence guarantee to optimal solutions in competitive Imperfect-Information Games (IIGs). The main reason is that the policy gradient theorem (Sutton et al., 1999) is established within the single agent situation, where the environment is Markovian. However, learning becomes non-stationary and non-Markovian when multiple agents learn simultaneously in a competitive environment.

An optimal solution to a 2-player zero-sum IIG usually refers to a Nash Equilibrium (NE), where no player could improve by unilaterally deviating to a different policy. Tremendous progress in computing NE solutions has been made by a family of tabular methods: Counterfactual regret

---

[*]Equal contribution. Correspondence to: Haobo Fu (haobofu@tencent.com).
[†]Work done while internship at Tencent AI Lab, Shenzhen, China.

minimization (CFR) (Zinkevich et al., 2008). CFR is a type of iterative self-play algorithm based on regret minimization, and it guarantees to converge to a NE with regard to the average policy in 2-player zero-sum IIGs. A perfect game model is required in CFR to sample many if not all actions from a state. To handle large-scale IIGs with CFR, abstractions (applied to either the action space or the state space) are usually employed to reduce the game to a manageable size (Moravčík et al., 2017; Brown & Sandholm, 2018; 2019)[1]. However, abstractions are domain specific (Waugh et al., 2009; Johanson et al., 2013; Ganzfried & Sandholm, 2014). More importantly, some large-scale IIGs are inherently difficult to be abstracted, such as the game of Mahjong (Li et al., 2020b).

In this paper, we investigate a large-scale IIG, i.e., 2-player (1-on-1) zero-sum Mahjong, whose information set size and game length are much larger than poker[2]. Li et al. (2020b) has recently developed a strong 4-player Mahjong agent based on supervised learning and traditional Reinforcement Learning (RL). In comparison, we study 1-on-1 Mahjong from a game-theoretic perspective, i.e., aiming for a NE. We are interested in methods using only trajectory samples to learn, as it is infeasible to consistently sample multiple actions for each state in large-scale IIGs with long episodes. We employ deep neural networks to generalize across states, since the state abstraction in 1-on-1 Mahjong is inherently difficult, as explained in the Appendix A.2. We make the following contributions.

- Inheriting the scalability of deep RL methods and the convergence property of CFR, we develop a new actor-critic algorithm, named Actor-Critic Hedge (ACH), for approaching a NE in large-scale 2-player zero-sum IIGs. ACH employs a deep neural network to approximate a type of weighted cumulative counterfactual regret. In the meantime, ACH minimizes the regret via generating self-play policies using Hedge (Freund & Schapire, 1997).

- We introduce a Neural-based Weighted CFR (NW-CFR), of which ACH is a practical implementation. We prove that the exploitability of the average policy in NW-CFR decreases at the rate of $O(T^{-1/2})$ under certain conditions, where $T$ denotes the number of iterations in NW-CFR.

- To facilitate research on large-scale 2-player zero-sum IIGs, we propose a 1-on-1 Mahjong benchmark. The corresponding game enjoys a large population in online games.

- We build a 1-on-1 Mahjong agent, named JueJong, based on ACH. In an initial evaluation against human players including a Mahjong champion[3], JueJong demonstrates superior performance.

## 2 NOTATIONS AND BACKGROUND

### 2.1 IMPERFECT-INFORMATION GAMES AND NASH EQUILIBRIUM

An IIG is usually described in an extensive-form game tree. A node (history) $h \in \mathcal{H}$ in the tree represents all information of the current situation. For each history $h$, there is a player $p \in \mathcal{P}$ or a chance player $c$ that should act at $h$. Define $P : \mathcal{H} \to \mathcal{P} \cup \{c\}$. When $P(h) \in \mathcal{P}$, the player $P(h)$ has to take an action $a \in \mathcal{A}(h)$, and $\mathcal{A}(h)$ is the set of legal actions in $h$. The chance player is responsible for taking actions for random events. The set of terminal nodes is denoted by $\mathcal{Z}$. For each player $p \in \mathcal{P}$, there is a payoff function defined on the set of terminal nodes, $u_p : \mathcal{Z} \to \mathbb{R}$. In this paper, we focus on 2-player zero-sum games, where $\mathcal{P} = \{0, 1\}$ and $u_0(z) + u_1(z) = 0$ for each $z \in \mathcal{Z}$.

For either player $p$, the set of histories $\mathcal{H}$ is partitioned into information sets (infosets). We denote the set of infosets for player $p$ by $\mathcal{I}_p$ and an infoset in $\mathcal{I}_p$ by $I_p$. Two histories $h, h' \in \mathcal{H}$ are in the same infoset if and only if $h$ and $h'$ are indistinguishable from the perspective of player $p$. Hence, a player $p$'s policy $\pi_p$ is defined as a function that maps an infoset to a probability distribution over legal actions. We further define $\mathcal{A}(I_p) = \mathcal{A}(h)$ and $P(I_p) = P(h)$ for any $h \in I_p$. A policy profile $\pi$ is a tuple of policies $(\pi_p, \pi_{-p})$, where $\pi_{-p}$ represents the player $p$'s opponent policy. The expected payoff for player $p$ under $\pi$ is denoted by $u_p(\pi_p, \pi_{-p})$. We use $\Delta(I)$ to denote the range of payoffs reachable from a history $h$ in infoset $I$. Let $\Delta = \max_{I \in \mathcal{I}_p, p \in \mathcal{P}} \Delta(I)$. $f^\pi(h)$ denotes the joint probability of reaching $h$ under $\pi$. $f_p^\pi(h)$ is the contribution of player $p$ to $f^\pi(h)$, and $f_{-p}^\pi(h)$ is the contribution of the opponent and chance: $f^\pi(h) = f_p^\pi(h) f_{-p}^\pi(h)$. We focus on the perfect-recall setting, where each

---

[1]DeepStack (Moravčík et al., 2017) employs sparse lookahead trees, much like the action abstraction.

[2]Without specification, we mean Heads-up No-Limit Texas Hold'em in this paper.

[3]Haihua Cheng is the Competition Mahjong champion of 2014 World Mahjong Master Tournament, 2018 Tencent Mahjong Tournament, and 2019 Tencent Mahjong Tournament.

player recalls the sequence of their own infosets reached. We define $f_p^\pi(I_p) = f_p^\pi(h), \forall h \in I_p$ and $f_{-p}^\pi(I_p) = \sum_{h \in I_p} f_{-p}^\pi(h)$. Hence, $f^\pi(I_p) = \sum_{h \in I_p} f^\pi(h) = f_p^\pi(I_p) f_{-p}^\pi(I_p)$.

A best response $BR(\pi_{-p})$ to $\pi_{-p}$ is a player $p$'s policy that satisfies $u_p(BR(\pi_{-p}), \pi_{-p}) = \max_{\pi_p'} u_p(\pi_p', \pi_{-p})$. A NE is a policy profile $\pi^*$, where each player plays a best response to the other: $u_p(\pi_p^*, \pi_{-p}^*) = \max_{\pi_p'} u_p(\pi_p', \pi_{-p}^*), \forall p \in \mathcal{P}$. The exploitability of a player's policy, denoted by $e(\pi_p)$, measures the performance gap between $\pi_p$ and a NE policy $\pi_p^*$: $e(\pi_p) = u_p(\pi_p^*, \pi_{-p}^*) - u_p(\pi_p, BR(\pi_p))$. The exploitability of $\pi$ is $\epsilon(\pi) = \frac{1}{|\mathcal{P}|} \sum_{p \in \mathcal{P}} e(\pi_p)$.

## 2.2 REINFORCEMENT LEARNING AND POLICY GRADIENT METHODS

RL usually assumes a Markov decision process, where the agent selects an action $a_i$ from the legal action set $\mathcal{A}(s_i)$ in state $s_i \in \mathcal{S}$[4] at each time step $i$. The agent then receives a reward $r_i$ from the environment and transitions to a new state $s_{i+1}$. The objective in RL is to learn a policy that maximizes the expected discounted returns, i.e., the state value, starting from any state $s$: $V^\pi(s) = \mathbb{E}^\pi[\sum_{j=i}^\infty \gamma^{j-i} r_j | s_i = s] = \mathbb{E}^\pi[G]$, with the discount factor $\gamma \in [0, 1)$.

Many methods in RL belong to policy gradient methods, in which the parameters $\theta$ of policy $\pi(a|s; \theta)$ are updated by performing gradient ascent directly on $\mathbb{E}^{\pi_\theta}[G]$. One early example is the standard REINFORCE algorithm (Williams, 1992) that updates $\theta$ in the direction $\nabla_\theta \log \pi(a|s; \theta) G$, which is an unbiased estimate of $\nabla_\theta \mathbb{E}^{\pi_\theta}[G]$. To further reduce the variance of the gradient, an action-independent baseline is often subtracted from $G$: $\nabla_\theta \log \pi(a|s; \theta)[G - B(s)]$. A recent policy gradient method, named Advantage Actor-Critic (A2C) (Mnih et al., 2016), learns a parameterized value function as the baseline: $B(s) = V(s; w)$, where $w$ often shares some parameters with $\theta$. Moreover, the value $G - B(s)$ is replaced in A2C with the estimated advantage of action $a$ in state $s$: $A(s, a) = Q(s, a) - V(s)$. The Q value $Q(s, a)$ is usually estimated using sampled rewards and predicted values of future states. A2C updates the parameters $\theta$ and $w$ in a synchronous manner.

In an asynchronous training environment, the behavioral policy is usually different from the learning policy. Proximal Policy Optimization (PPO) (Schulman et al., 2017) takes this discrepancy into account by multiplying A2C gradient with an importance ratio $r(a|s; \theta) = \pi(a|s; \theta)/\pi(a|s; \theta_{old})$, which results in $r(a|s; \theta) \nabla_\theta \log \pi(a|s; \theta) A(s, a) = \nabla_\theta r(a|s; \theta) A(s, a)$. Furthermore, PPO constrains the KL divergence between the learning policy $\pi_\theta$ and the old behavioral policy $\pi_{\theta_{old}}$ by clipping the ratio to a small interval around 1.0.

## 2.3 COUNTERFACTUAL REGRET MINIMIZATION

CFR is an iterative algorithm that minimizes the total regret of policy by minimizing the cumulative counterfactual regret in every state (infoset). The cumulative counterfactual regret of an action $a$ in state $s$ is defined as $R_t^c(s, a) = \sum_{k=1}^t r_k^c(s, a)$, where $r_k^c(s, a)$ denotes the instantaneous counterfactual regret of action $a$ in state $s$ at iteration $k$. $r_t^c(s, a)$ is equal to $f_{-p}^{\pi_t}(s) A^{\pi_t}(s, a)$ (as proven in Srinivasan et al. (2018)), where $A^{\pi_t}(s, a)$ is the advantage function of player $p = P(s)$ at state $s$ as defined in traditional RL. Intuitively, $r_t^c(s, a)$ represents how regretful the player $p$ is that he does not select the action $a$ in state $s$ at iteration $t$. The term $f_{-p}^{\pi_t}(s) = \sum_{h \in s} f_{-p}^{\pi_t}(h)$ is needed here to reflect the fact that reaching state $s$ is also controlled by the opponent $-p$ and chance. To minimize the regret, a regret minimizer is utilized in each state at each iteration, generating a series of local policies $\pi_1(s), \ldots, \pi_t(s)$. Once $\pi_t$ is obtained, $r_t^c(s, a)$ is generated by a game tree traversing using $\pi_t$, and $R_t^c(s, a)$ is updated accordingly: $R_t^c(s, a) = R_{t-1}^c(s, a) + r_t^c(s, a)$.

For either player $p$, define the total regret as $R_T = \max_{\pi_p'} \sum_{t=1}^T (u_p(\pi_p', \pi_{-p,t}) - u_p(\pi_{p,t}, \pi_{-p,t}))$. It is proven in Zinkevich et al. (2008) that $R_T \leq \sum_{s \in \mathcal{S}} [R_T^c(s)]^+$, where $[\cdot]^+ = \max\{\cdot, 0\}$ and $R_T^c(s) = \max_{a \in \mathcal{A}(s)} R_T^c(s, a)$. As a result, the total regret can be minimized by minimizing the cumulative counterfactual regret at each state. Moreover, the average policy $\bar{\pi}_T(a|s) = \sum_{t \in T} f_p^{\pi_t}(s) \pi_t(a|s) / \sum_{t \in T} f_p^{\pi_t}(s)$ for both players converges to a NE if $R_T$ for both players grows sub-linearly with $T$ (Zinkevich et al., 2008). There are two commonly used regret minimizers: Regret Matching (RM) (Hart & Mas-Colell, 2000) and Hedge (Cesa-Bianchi & Lugosi, 2006). In RM, a

---

[4]We use state $s$ and infoset $I_p$ interchangeably in this paper.

player selects an action with probability in proportion to its positive cumulative counterfactual regret. In Hedge, the policy $\pi_{t+1}(a|s)$ is decided according to:

$$\pi_{t+1}(a|s) = \frac{e^{\eta(s)R_t^c(s,a)}}{\sum_{a'} e^{\eta(s)R_t^c(s,a')}}. \tag{1}$$

If the player plays according to Hedge in state $s$ and $\eta(s) = \sqrt{8\log(|\mathcal{A}(s)|)/(\Delta^2(s)T)}$, $R_T^c(s) \leq \Delta(s)\sqrt{\log(|\mathcal{A}(s)|)T/2}$ (Cesa-Bianchi & Lugosi, 2006). In other words, the total regret grows sub-linearly with $T$, and therefore the average policy $\bar{\pi}_T(a|s)$ for both players converges to a NE.

## 3 THE MOTIVATION OF ACTOR-CRITIC HEDGE

In this section, we motivate ACH by introducing a new neural-based CFR algorithm, NW-CFR, which employs a neural network to generalize across states and relies on only trajectory samples for training. The key idea in NW-CFR is that a neural network (called the policy net) is used to approximate the expectation of the sum of sampled advantages $R_t^a(s,a) := \mathbb{E}[\sum_{k=1}^{t} \tilde{A}^{\pi_k}(s,a)]$. $\tilde{A}^{\pi_k}(s,a)$ is set to $A^{\pi_k}(s,a)$ if $s$ is visited at iteration $k$ (given that only one trajectory is sampled at each iteration) and 0 otherwise. As a result, the expectation $\mathbb{E}[\tilde{A}^{\pi_k}(s,a)]$ depends on both the advantage $A^{\pi_k}(s,a)$ and the sampling policy $\mu_k$ at iteration $k$.

At the beginning of iteration $t$, suppose $R_{t-1}^a(s,a)$ is well approximated by the output $y(a|s;\theta_{t-1})$ of the policy net, parameterized as $\theta_{t-1}$ in NW-CFR. The policy $\pi_t$ for iteration $t$ is first obtained via Hedge, i.e., softmaxing on $\eta(s)y(a|s;\theta_{t-1})$. The reason we use Hedge instead of RM is that softmaxing is shift-invariant, which may be more robust to the function approximation error compared to the threshold operation in RM. $M$ trajectories are then sampled into a buffer $\mathcal{B}_v$ via self-play using the policy profile $\pi_t = (\pi_{p,t}, \pi_{-p,t})$. Those samples in $\mathcal{B}_v$ are used to train the action value net, parameterized as $\omega$, by minimizing the squared loss: $\frac{1}{2}[Q(s,a;\omega) - G]^2$, where $G$ is the sampled return. Afterwards, another $M$ trajectories are sampled into a buffer $\mathcal{B}_\pi$ via self-play using another sampling policy profile $\mu_t = (\mu_{p,t}, \pi_{-p,t})$. We use an additional behavioral policy $\mu_{p,t}$ for the player $p$ for more flexibility in the convergence (more details are provided in the next section). The policy $\theta$ is then optimized according to the loss function $\mathcal{L}_\pi = \sum_{s \in \mathcal{S}} \mathcal{L}_\pi(s)$, where

$$\mathcal{L}_\pi(s) = \begin{cases} \sum_a \{y(a|s;\theta) - [y(a|s;\theta_{t-1}) + \frac{1}{M}\sum_{i=1}^{M} \mathbb{1}_{s \in \tau_i} A^{\pi_t}(s,a)]\}^2 & \text{if } s \in \mathcal{B}_\pi, \\ \sum_a \{y(a|s;\theta) - [y(a|s;\theta_{t-1}) + 0]\}^2 & \text{otherwise,} \end{cases} \tag{2}$$

where $\tau_i$ is the $i$th sampled trajectory in $\mathcal{B}_\pi$. The next iteration of NW-CFR begins after the policy training finishes. The pseudocode of NW-CFR is given in Algorithm 1. Note that we present NW-CFR from the perspective of player $p$, and the same procedure applies to the player $-p$. NW-CFR runs concurrently for both players and synchronizes at the beginning of each iteration.

A potential benefit of training on the sampled advantage over the sampled instantaneous counterfactual regret (as done in Brown et al. (2019), Li et al. (2020a), and Steinberger et al. (2020)) is that the variance of the sampled advantage could be much lower. As $r_k^c(s,a) = f_{-p}^{\pi_k}(s)A^{\pi_k}(s,a)$ (Srinivasan et al., 2018), the sampled instantaneous counterfactual regret $\tilde{r}_k^c(s,a)$ is $[f^{\mu_k}(s)]^{-1}f_{-p}^{\pi_k}(s)\tilde{A}^{\pi_k}(s,a) = [f_p^{\mu_k}(s)]^{-1}\tilde{A}^{\pi_k}(s,a)$, where $f_p^{\mu_k}(s)$ is the probability of reaching $s$ considering only the contribution of $\mu_{p,k}$. The larger variance (due to $[f_p^{\mu_k}(s)]^{-1}$) of $\tilde{r}_k^c(s,a)$ may have a negative influence on the performance when function approximation is used with only trajectory samples. This influence is magnified in large-scale games, as $[f_p^{\mu_k}(s)]^{-1}$ becomes large.

## 4 THEORETICAL PROPERTIES OF NW-CFR

In this section, we prove the convergence of NW-CFR to an approximate NE. To this end, we first deduce that the policy target in NW-CFR is $R_t^a(s,a)$, which is a type of weighted cumulative counterfactual regret. We then define a new family of CFR algorithms: weighted CFR. Finally, we show that NW-CFR is equivalent to a type of weighted CFR with Hedge under certain conditions. The convergence properties of weighted CFR with Hedge and hence NW-CFR are proven accordingly.

We can rewrite the NW-CFR policy loss (in Equation 2) at iteration $t$ as $\mathcal{L}_\pi = \sum_{s \in \mathcal{S}} \sum_a \{y(a|s;\theta) - [y(a|s;\theta_{t-1}) + \frac{1}{M}\sum_{i=1}^{M} \mathbb{1}_{s \in \tau_i} A^{\pi_t}(s,a)]\}^2$. In other words, the target of $y(a|s;\theta)$ at iteration $t$ is

---

**Algorithm 1:** Neural-based Weighted CFR

---

**Function** `Neural-based Weighted CFR`($p, T, M, \theta, \omega$)**:**

$\quad$ $\omega_0 \leftarrow \omega, \theta_0 \leftarrow \theta$.

$\quad$ **for** $t \leftarrow 1$ **to** $T$ **do**

$\quad\quad$ # **Obtain the policy**: $\pi_t \leftarrow \text{Softmax}(\eta(s)y(a|s; \theta_{t-1}))$.

$\quad\quad$ # **Train the action value net**:

$\quad\quad$ Reset $\mathcal{B}_v \leftarrow \emptyset$.

$\quad\quad$ **for** $i \leftarrow 1$ **to** $M$ **do**

$\quad\quad\quad$ $\lfloor$ $\tau_i \sim \text{SelfPlay}(\pi_{p,t}, \pi_{-p,t}), \mathcal{B}_v \leftarrow \mathcal{B}_v \cup \tau_i$

$\quad\quad$ Train $\omega$ on the loss $\mathbb{E}_{(s,a,G) \sim \mathcal{B}_v}\{\frac{1}{2}[Q(s,a;\omega) - G]^2\}$

$\quad\quad$ $\omega_t \leftarrow \omega$

$\quad\quad$ # **Train the policy net**:

$\quad\quad$ Reset $\mathcal{B}_\pi \leftarrow \emptyset$.

$\quad\quad$ **for** $i \leftarrow 1$ **to** $M$ **do**

$\quad\quad\quad$ $\lfloor$ $\tau_i \sim \text{SelfPlay}(\mu_{p,t}, \pi_{-p,t}), \mathcal{B}_\pi \leftarrow \mathcal{B}_\pi \cup \tau_i$

$\quad\quad$ Estimate $A^{\pi_t}(s,a), \forall a \in \mathcal{A}(s), \forall s \in \mathcal{B}_\pi$ as $Q(s,a;\omega_t) - \sum_b \pi_t(b|s)Q(s,b;\omega_t)$.

$\quad\quad$ Sum aggregate advantage $A^{\pi_t}(s,a)$ in $\mathcal{B}_\pi$ by state $s$.

$\quad\quad$ Train $\theta$ on the loss $\mathbb{E}_{s\sim\mathcal{S}}\{\mathcal{L}_\pi(s)\}$, where $\mathcal{L}_\pi(s)$ is defined in Equation 2.

$\quad\quad$ $\theta_t \leftarrow \theta$. # Save $\theta_t$ to a policy buffer.

---

$y(a|s; \theta_{t-1}) + \frac{1}{M}\sum_{i=1}^{M} \mathbb{1}_{s\in\tau_i} A^{\pi_t}(s,a)$, whose expectation is

$$\mathbb{E}_{\tau_{k,i}\sim(\mu_{p,k},\pi_{-p,k})}\left[\sum_{k=1}^{t}\sum_{i=1}^{M}\frac{1}{M}\mathbb{1}_{s\in\tau_{k,i}}A^{\pi_k}(s,a)\right] = \mathbb{E}_{\tau_k\sim(\mu_{p,k},\pi_{-p,k})}\left[\sum_{k=1}^{t}\tilde{A}^{\pi_k}(s,a)\right] = R_t^a(s,a).$$

Also, $\mathbb{E}_{\tau_k\sim(\mu_{p,k},\pi_{-p,k})}\mathbb{1}_{s\in\tau_k}A^{\pi_k}(s,a) = f_p^{\mu_k}(s)f_{-p}^{\pi_k}(s)A^{\pi_k}(s,a)$, where $f_p^{\mu_k}(s)f_{-p}^{\pi_k}(s)$ is the reaching probability of state $s$ at iteration $k$. Therefore, $R_t^a(s,a) = \sum_{k=1}^{t} f_p^{\mu_k}(s)f_{-p}^{\pi_k}(s)A^{\pi_k}(s,a)$. Since $r_k^c(s,a) = f_{-p}^{\pi_k}(s)A^{\pi_k}(s,a)$ (Srinivasan et al., 2018), we have $R_t^a(s,a) = \sum_{k=1}^{t} f_p^{\mu_k}(s)r_k^c(s,a)$.

As a result, $R_t^a(s,a)$ can be viewed as a type of weighted cumulative counterfactual regret that multiples each instantaneous counterfactual regret with $f_p^{\mu_k}(s)$. Without loss of generality, we define a new family of CFR algorithms, weighted CFR, below. Afterwards, we present Theorem 1.

**Definition 1.** *Weighted CFR follows the same procedure as the original CFR (Zinkevich et al., 2008), except that the instantaneous counterfactual regret $r_t^c(s,a)$ is weighted by some weight $w_t(s)$, $w_t(s) > 0$ and $\sum_{t=0}^{\infty} w_t(s) = \infty$. The original CFR is a type of weighted CFR with $w_t(s) = 1.0$.*

**Theorem 1.** *NW-CFR is equivalent to a type of weighted CFR with Hedge when $w_t(s) = f_p^{\mu_t}(s) > 0$, given that enough trajectories are sampled and $y(a|s; \theta_t)$ is sufficiently close to $R_t^a(s,a)$. Further, if $\eta(s) = \sqrt{8\ln|\mathcal{A}(s)|/\{[w_h(s)]^2\Delta^2(s)T\}}$ and $w_t(s) = f_p^{\mu_t}(s) \in [w_l(s), w_h(s)] \subset (0,1], t = 1,\ldots,T$, the average policy[5] $\bar{\pi}$ of the corresponding weighted CFR with Hedge and equivalently NW-CFR with $\bar{\pi}_p(a|s) = \sum_{t=1}^{T}[f_p^{\pi_t}(s)\pi_t(a|s)]/\sum_{t=1}^{T}f_p^{\pi_t}(s), \forall p \in \mathcal{P}$, has $\epsilon$ exploitability after $T$ iterations, where*

$$\epsilon \leq |\mathcal{S}|\Delta\sqrt{\frac{1}{2T}\ln|\mathcal{A}|} + \Delta\sum_{s\in\mathcal{S}}\frac{w_h(s) - w_l(s)}{w_h(s)}. \tag{3}$$

In Theorem 1, we proved that the exploitability $\epsilon$ of NW-CFR is bounded by Equation 3 when $y(a|s; \theta_t)$ is sufficiently close to $R_t^a(s,a)$. There are two terms in the bound. The first term converges to zero at the rate of $O(T^{-1/2})$, and the second term is $O(1)$, which is weighted by the sum of the normalized range of the weights $\sum_{s\in\mathcal{S}}(w_h(s) - w_l(s))/w_h(s)$. However, it is possible to reduce the second term to an arbitrarily small value via tightening the range of $f_p^{\mu_t}(s)$, which is experimentally demonstrated in the Appendix D.

---

[5]Given $\pi_{p,t}$ at each iteration, we could obtain $\bar{\pi}_p$ using the techniques introduced in Steinberger (2019).

**Corollary 1.** *If the behavioral policy $\mu_{p,t}$ for each player $p \in \mathcal{P}$ is constant across iterations, and $f_p^{\mu_t}(s) > 0, \forall s \in \mathcal{S}, t > 0$, NW-CFR is equivalent to CFR with Hedge when $y(a|s; \theta_t)$ is sufficiently close to $R_t^a(s, a)$.*

As shown in Corollary 1, when the behavioral policy $\mu_{p,t}$ for each player $p \in \mathcal{P}$ is time-invariant, i.e., $w_h(s) = f_p^{\mu_t}(s) = w_l(s), \forall s \in \mathcal{S}, t > 0$, the second term of $\epsilon$ in Equation 3 vanishes, and CFR with Hedge is recovered. All the proofs are given in the Appendix C.

## 5 ACH: A Practical Implementation of NW-CFR

When applying NW-CFR to large-scale problems, two practical issues need to be addressed:

**The average policy.** Theorem 1 and Corollary 1 state the convergence property of the average policy in NW-CFR. Yet, as pointed out in Srinivasan et al. (2018), Hennes et al. (2020), and Perolat et al. (2021), obtaining the average policy with deep neural nets in large-scale games is inherently difficult, due to either the computation or the memory demand. Alternatively, we could employ some additional technique to hopefully induce the current policy convergence towards a NE. Srinivasan et al. (2018) and Hennes et al. (2020) handled this by adding an entropy regularization to the current policy training, which is, to some extent, theoretically justified later in Perolat et al. (2021).

**Training on states not sampled.** Theoretically, in order to optimize Equation 2, we need to collect both sampled and non-sampled states. Optimizing with only sampled states makes $y(a|s; \theta_t)$ a biased estimation of $R_t^a(s, a)$. Yet, collecting non-sampled states may be intractable in large-scale games (Li et al., 2020a) or in situations where a perfect environment model is not available.

To strike a balance between theoretical soundness and practical efficiency, we provide a practical implementation of NW-CFR, which is ACH. ACH adapts NW-CFR by training the current policy with an entropy regularization on only sampled states, without the calculation of the average policy. In order to utilize distributed clusters, ACH employs a framework of decoupled acting and learning (similar to IMPALA (Espeholt et al., 2018)), trains the network with mini-batches, and handles asynchronous training with the importance ratio clipping of PPO. The behavior policy $\mu_{p,t}$ is set to $\pi_{p,t}$[6] in ACH. More details of ACH are presented in the Appendix E.

## 6 Related Work

To obviate the need of abstractions, various neural forms of CFR methods have been developed. An early work of this direction is regression CFR (Waugh et al., 2015), which calculates weights for a number of hand-crafted features to approximate the regret. Deep CFR (Brown et al., 2019) is similar to regression CFR but employs a neural network to approximate the regret. Also, deep CFR traverses a part of the game tree using external sampling, in comparison to the full traversal of the game tree in regression CFR. Double neural CFR (Li et al., 2020a) is another method that approximates the regret and the average policy using deep neural networks, where a novel robust sampling technique is developed. Both deep CFR and double neural CFR build on the tabular Monte Carlo CFR (MCCFR) (Lanctot et al., 2009), where either outcome sampling (sampling one action in a state) or external sampling (sampling all actions in a state) could be employed.

When dealing with games with long episodes, a necessity may be that only trajectory samples are allowed. To improve the learning performance with only trajectory samples, DREAM (Steinberger et al., 2020) adapts deep CFR by using a learned Q-baseline, which is inspired by the variance reduction techniques in tabular MCCFR (Schmid et al., 2019; Davis et al., 2020). Another recent work using only trajectory samples is ARMAC (Gruslys et al., 2020). By replaying through past policies and using a history-based critic, ARMAC predicts conditional advantages, based on which the policy for each iteration is generated. Other popular neural network based methods, which learn from trajectory samples and are inspired by game-theoretic approaches other than CFR, include neural fictitious self-play (Heinrich & Silver, 2016), policy space response oracles (Lanctot et al., 2017), and exploitability descent (Lockhart et al., 2019), all of which require to compute an approximate best response at each iteration. Such computation may be prohibitive in large-scale games.

---

[6]In a preliminary experiment presented in the Appendix F, we find that the performance of the current policy in ACH (trained with an entropy regularization) is not sensitive to the choice of $\mu_{p,t}$.

The most related methods to ACH are Regret Policy Gradient (RPG) (Srinivasan et al., 2018) and Neural Replicator Dynamics (NeuRD) (Hennes et al., 2020), both of which employ the actor-critic framework and thus have similar computation and memory complexities as ACH. RPG minimizes a loss that is an upper bound on the regret after threshold, and the corresponding policy gradient is $\nabla_\theta^{RPG}(s) = -\sum_a \nabla_\theta [Q(s,a;w) - \sum_b \pi(b|s;\theta)Q(s,b;w)]^+$. However, RPG requires an $l_2$ projection after every gradient step for the convergence to a NE, while such projection is not required in ACH. NeuRD is inspired by the replicator dynamics, a well studied model in evolutionary game theory (Gatti et al., 2013). The policy gradient in NeuRD is $\nabla_\theta^{NeuRD}(s) = \sum_a [\nabla_\theta y(a|s;\theta)][Q(s,a;w) - \sum_b \pi(b|s)Q(s,b;w)]$, where $y(a|s;\theta)$ is the output of the policy net. There are important differences between ACH and NeuRD in how the algorithm is motivated and how the policy net at each iteration is optimized. Also, the convergence analysis is given only for the single-state all-actions tabular NeuRD (Hennes et al., 2020). Yet, we prove the convergence of NW-CFR, of which ACH is a practical implementation, in full extensive-form games.

## 7 EXPERIMENTAL STUDIES

We firstly introduce a 1-on-1 Mahjong benchmark, on which we compare ACH with related state-of-the-art methods of similar computation complexity: PPO, RPG, and NeuRD. Since our goal is to approximate a NE, the standard and default performance metric, exploitability, is employed. We approximate a lower bound on the exploitability of an agent by training a best response against it as suggested in Timbers et al. (2020) and Steinberger et al. (2020), because traversing the full game tree to compute the exact exploitability is intractable in such a large-scale game as 1-on-1 Mahjong. As a complement, head-to-head performance of different methods on 1-on-1 Mahjong is also presented. Moreover, the agent obtained by ACH is evaluated against practised Mahjong human players. To further validate the performance of ACH in IIGs other than 1-on-1 Mahjong, experimental results on a non-trivial poker game, i.e., heads-up Flop Hold'em Poker (FHP) (Brown et al., 2019) are presented. Deep CFR with outcome sampling (OS-DCFR) and DREAM are added to enable a more thorough comparison. Additional results on smaller benchmarks from OpenSpiel (Lanctot et al., 2019) are given in the Appendix G. Note that results are reported for the current policy (of ACH, PPO, RPG, and NeuRD) and the average policy (of OS-DCFR and DREAM) respectively.

### 7.1 A 2-PLAYER ZERO-SUM MAHJONG BENCHMARK

Mahjong is a tile-based game that is played world wide with many regional variations, such as Japanese Riichi Mahjong and Competition Mahjong. Like poker, Mahjong is an IIG and is full of strategy, chance, and calculation. To facilitate Mahjong research from a game-theoretic perspective, we propose a 2-player zero-sum Mahjong benchmark, whose game rules are similar to Competition Mahjong. The corresponding game, "2-player Mahjong Master", is played by humans in Tencent mobile games. A full description of the game rules is in the Appendix A.1. Apart from being the first benchmark for the 1-on-1 Mahjong game, our benchmark has a larger infoset size and a longer game length (the effects are explained in the Appendix A.3), compared with existing poker benchmarks (Lanctot et al., 2019). The infoset size (i.e., the number of distinct histories in an infoset) in 1-on-1 Mahjong is around $10^{11}$, compared to $10^3$ in poker. This is due to the fact that only two private cards are invisible in poker, while there are 13 invisible tiles in 1-on-1 Mahjong. In addition, players can decide up to about 40 sequential actions in 1-on-1 Mahjong, whereas most 1-on-1 poker games end within 10 steps. More details about the 1-on-1 Mahjong benchmark are given in the Appendix A.

### 7.2 RESULTS ON OUR 1-ON-1 MAHJONG BENCHMARK

All methods run in an asynchronous training platform with overall 800 CPUs, 3200 GB memory, and 8 M40 GPUs in the Ubuntu 16.04 operating system. Each method shares the same neural network architecture, a full description of which is given in the Appendix B. We performed a mild hyper-parameter search on PPO and shared the best setting for all methods. The advantage value is estimated by the Generalized Advantage Estimator (GAE($\lambda$)) (Schulman et al., 2016) for all methods. An overview of the hyper-parameters is listed in the Appendix H.1.

**Approximate Lower Bound Exploitability**. To approximate a lower bound on the exploitability of the agents obtained by each method, we train a best response against each agent. The agent of

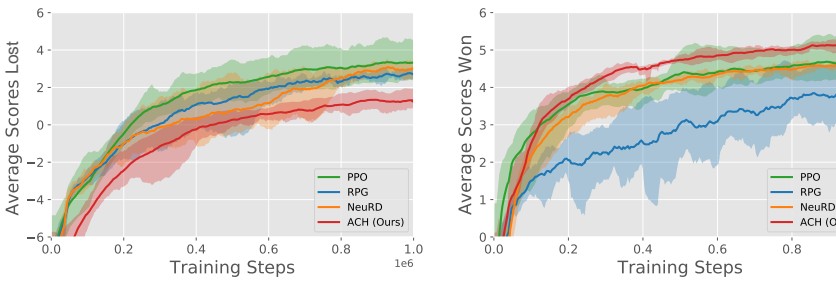

(a) Approximate Lower Bound Exploitability       (b) Head-to-Head performance

Figure 1: **(a)**: The training curves of the best response against each agent. **Lower is better. (b)**: The training curves of each agent. The performance of an agent is evaluated by the average scores the agent wins against a common rule-based agent. **Higher is better**. We report the mean as solid curves and the range of the average scores across 5 independent runs as shaded regions.

each method is selected at the $1e6$th training step. Note that the agent is fixed as a part of the 1-on-1 Mahjong environment when training the best response. We train the best response using PPO with the same hyper-parameters that were used to train the PPO agent with self-play. During the training of the best response, we evaluate the best response every 500 training steps using 10,000 head-to-head plays against each agent. According to the average scores each agent loses to its best response in Figure 1(a), we may conclude that ACH is significantly more difficult to exploit than other methods in the large-scale 1-on-1 Mahjong environment.

**Head-to-Head Evaluation**. We then compare the head-to-head performance of PPO, RPG, NeuRD, and ACH in the 1-on-1 Mahjong environment. First, we compare the training process of each method by evaluating each agent every 500 training steps against a common rule-based agent[7] using 10,000 head-to-head plays, the results of which are shown in Figure 1(b). As we can see, all methods beat the common rule-based agent significantly, while ACH has a clear advantage over other methods in terms of stability and final performance. The relatively slow convergence of RPG may due to the threshold operation on the advantage, which could reduce sample efficiency in large-scale IIGs. Second, the superior performance of ACH is further validated in head-to-head evaluations with other agents in Table 1, where all the agents are selected at the $1e6$th training step. The agent of ACH wins all other agents by a significant margin.

|  | PPO | RPG | NeuRD |
|---|---|---|---|
| RPG | **-0.21** $\pm 0.05$ | - | - |
| NeuRD | **-0.03** $\pm 0.02$ | **0.04** $\pm 0.10$ | - |
| ACH | **0.39** $\pm 0.02$ | **0.66** $\pm 0.05$ | **0.41** $\pm 0.07$ |

Table 1: Mean ($\pm$ standard deviation) of the average winning scores of the row agents against the column agents. The statistics are estimated by 5 independent runs (resulting 5 different agents for each method). In each run, the average winning scores are obtained via 10,000 head-to-head plays.

**Human Evaluation**. We evaluate the agent, selected at the $1e6$th training step, of ACH against human players. First, the agent, named JueJong, is roughly evaluated by playing over 7,700 games against 157 practiced Mahjong players, where JueJong won an average of 4.56 scores per game. Second, we select the top 4 out of 157 players according to their performances against JueJong and play JueJong against the four players for 200 games each. As shown in Figure 2(a), the average winning scores of JueJong oscillate in the first 120 games but all plateau above 0 afterwards. More importantly, we evaluate JueJong against the Mahjong champion Haihua Cheng for 1,000 games, as shown in Figure 2(b). After playing 1,000 games, JueJong won the champion by a score of $0.82 \pm 0.96$ (mean $\pm$ standard deviation), with a p-value of 0.19 under one-tailed t-test. Hence, we may conclude that Haihua Cheng failed to exploit JueJong effectively within 1,000 games.

---

[7]The rule-based agent is implemented such that it selects the action Hu, Ting, Kong, Chow, and Pong in descending priority whenever available and discards the tile that has the fewest neighbours.

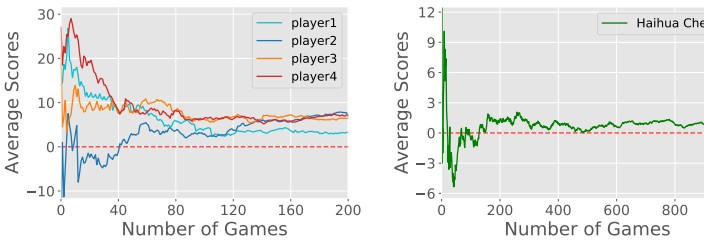

(a) Against 4 practiced Mahjong players.   (b) Against the Mahjong champion.

Figure 2: Performance of JueJong against human players.

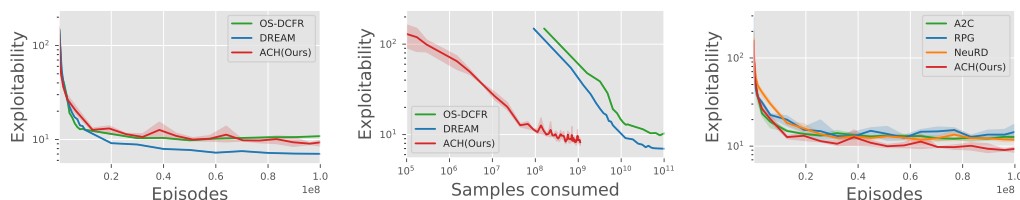

Figure 3: The exploitability on FHP, with the x-axis being the number of episodes generated (**left** and **right**) and the number of samples consumed (**middle**). We report the mean as solid curves and the range as shaded regions across 3 independent runs. OS-DCFR and DREAM were run once as their performances are relatively stable, according to Brown et al. (2019) and Steinberger et al. (2020).

### 7.3    RESULTS ON THE FHP BENCHMARK

We further evaluate ACH and compare it with OS-DCFR, DREAM, A2C[8], RPG, and NeuRD on FHP. FHP is a simplified Heads-up Limit Texas Hold'em (HULH), which includes only the first two of the four bettings in HULH. It is a medium-sized game with over $10^{12}$ nodes and $10^9$ infosets. All the methods share the same neural network architecture proposed in Brown et al. (2019). We perform a mild hyper-parameter search for ACH, A2C, RPG, and NeuRD. For OS-DCFR and DREAM, we follow the hyper-parameters presented in Steinberger et al. (2020). The exploitability is measured in the number of chips per game, where a big blind is 100 chips. All the hyper-parameters and the running environment are described in the Appendix H.2.

As shown in Figure 3, ACH performs competitively with OS-DCFR and slightly worse than DREAM on FHP in terms of exploitability per episodes generated. However, ACH is much more training efficient: ACH achieves an exploitability of 10 almost **100** times faster than DREAM and **1,000** times faster than OS-DCFR. Also, in comparison with methods of similar training complexity (A2C, RPG, and NeuRD), ACH converges significantly faster and achieves a lower exploitability.

## 8    CONCLUSIONS

In this paper, we investigated the problem of adapting policy gradient methods in deep RL to tackle a large-scale IIG, i.e., 1-on-1 Mahjong. To this end, we developed a new model-free actor-critic algorithm, i.e., ACH, for approximating a NE in large-scale IIGs. ACH is memory and computation efficient, as it uses only trajectory samples at the current iteration and requires no computation of best response. ACH is theoretically justified as it is derived from a new neural-based CFR, i.e., NW-CFR, of which we proved the convergence to an approximate NE in 2-player zero-sum IIGs under certain conditions. The superior performance of ACH was validated on both our 1-on-1 Mahjong benchmark and other common benchmarks. Secondly, to facilitate research on large-scale IIGs, we proposed the first 1-on-1 zero-sum Mahjong benchmark, whose infoset size and game length are much larger than poker. Finally, using ACH we obtained the 1-on-1 Mahjong agent JueJong, which has demonstrated stronger performance against the Mahjong champion Haihua Cheng.

---

[8]PPO is replaced with A2C in a synchronous training environment.

## ACKNOWLEDGEMENT

We thank the Mahjong champion Haihua Cheng for his efforts in this work. We appreciate the support from Tencent Mahjong (https://majiang.qq.com). We are grateful to Tencent AI Arena (https://aiarena.tencent.com) for providing the powerful computing capability to the experiments on 1-on-1 Mahjong.

## REPRODUCIBILITY STATEMENT

The experiments on 1-on-1 Mahjong were run in a large cluster of thousands of machines, on which we have developed an efficient actor-learner training platform, similar to IMPALA (Espeholt et al., 2018). The code of the platform is not released currently but is planned to be open sourced in the near future. The code of the 1-on-1 Mahjong benchmark is available at https://github.com/yata0/Mahjong. The code of ACH is available at https://github.com/Liuweiming/ACH_poker. All the hyper-parameters for all the experiments are listed in the Appendix H. All the theoretical results are presented in the main text, with all the proofs given in the Appendix C.

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

## A  INTRODUCTION OF THE 1-ON-1 MAHJONG BENCHMARK

Mahjong is a tile-based game that is played world wide with many regional variations, such as Japanese Riichi Mahjong and Competition Mahjong. Like poker, Mahjong is an IIG and is full of strategy, chance, and calculation. In this paper, we investigate a 1-on-1 Mahjong game, whose information set size is much larger than poker, as shown in Figure 4. The game rules of 1-on-1 Mahjong are similar to Competition Mahjong. The corresponding game, "2-player Mahjong Master", is played by humans in Tencent mobile games (https://majiang.qq.com).

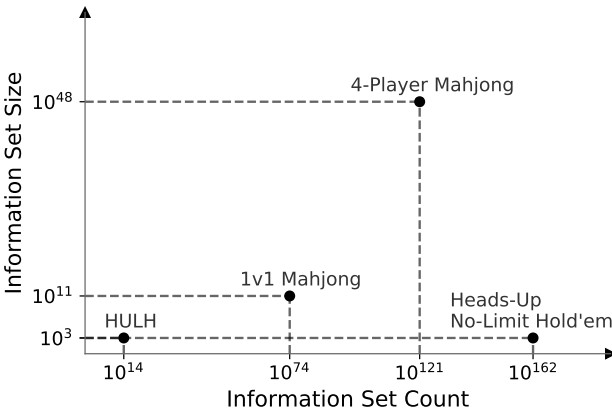

Figure 4: The game complexity of Heads-up Limit Texas Hold'em (HULH), Heads-up No-Limit Texas Hold'em, 1-on-1 Mahjong, and 4-Player Mahjong.

### A.1  THE GAME RULES

| Category | Name | Type | Copy | Count |
|----------|------|------|------|-------|
| Simples | Characters | 一 二 三 四 五 六 七 八 九 | 4 | 36 |
| Honors | Winds | 東 南 西 北 | 4 | 16 |
| Honors | Dragons | 中 發 □ | 4 | 12 |
| Bonus | Flowers | 梅 蘭 竹 菊 | 1 | 4 |
| Bonus | Seasons | 春 夏 秋 冬 | 1 | 4 |

Figure 5: A list of all tiles in the 1-on-1 Mahjong game.

There are 24 unique tiles and 72 tiles in total in the 1-on-1 Mahjong game, as shown in Figure 5. At the beginning of the 1-on-1 Mahjong game, each player is dealt with 13 tiles, the content of which is invisible to the other. Afterwards, each player takes actions in turn. Typically, the first player draws a tile from the deck wall and then discards a tile, and the next player takes the same types of actions in sequence. There are exceptional cases where the player does not draw a tile from the deck wall but Chow, Pong, or Kong the tile the opponent just discarded. Afterwards, the player discards a tile, and the game proceeds. Also, after drawing a tile from the deck wall, there are cases where the player could Kong, after which the player draws and discards a tile in sequence. There are 10 types of actions with 105 different actions in total, a full description of which is listed in Table 2.

| Type | Description | #Action |
|---|---|---|
| Discard | Discard one of the tiles in the hand. | 16 |
| Pong | Make a meld of 3 identical tiles by seizing the opponent's latest discarded tile. | 16 |
| Concealed-Kong | Make a meld of 4 identical tiles when they are in the hand, and the Concealed-Kong is not revealed to the opponent. | 16 |
| Kong | Make a meld of 4 identical tiles by seizing the opponent's latest discarded tile. | 16 |
| Add-Kong | Make a meld of 4 identical tiles by adding a tile to his own exposed Pong. | 16 |
| Chow | Make a meld of 3 character tiles in a row by seizing the opponent's latest discarded tile. | 21 |
| Draw | Give up the action of Pong, Chow, or Kong, and draw a tile from the deck wall. | 1 |
| Ting | When there is only one tile away from a legal hand, the player can declare Ting. | 1 |
| Hu | Form a legal hand by drawing a tile or seizing the opponent's latest discarded tile. The Hu action ends the game. | 1 |
| Pass-Hu | Give up the action of Hu. Afterwards, the player should select another legal action. | 1 |

Table 2: Different actions in the 1-on-1 Mahjong game.

The goal of each player is to complete a legal hand prior to the opponent, by drawing a tile or using the tile the opponent just discarded. A legal hand is generally in the form of four melds and a pair, with an exception of 7 pairs. Different categories of legal hands, 64 in total, come with different points, which is fully described in Table 3. Besides, a legal hand can belong to multiple categories, and the score of a legal hand is the sum of points of corresponding categories. Legal hands with higher points are generally more difficult to form in terms of either luck or strategy, and it is critical for a player to trade off the winning probability and the corresponding winning points. If no player completes a legal hand before the tiles are exhausted, the game is tied. A flow chart of the game is illustrated in Figure 6.

| | Category name | Points | Pattern descriptions | Conflicts |
|---|---|---|---|---|
| 1 | Pure Double Chow | 1 | Two identical sequences of 3 character tiles. | |
| 2 | Short Straight | 1 | A hand with two successive sequences. | |
| 3 | Two Terminal Chows | 1 | A hand with two sequences 123 and 789. | |
| 4 | Melded Kong | 1 | A hand with exposed quads. | |
| 5 | Edge Wait | 1 | A hand completion with the situation there is only one tile name to complete because you have the incomplete sequence in edge (12 or 89). | |
| 6 | Closed Wait | 1 | A hand completion with the situation there is only one tile name to complete because you have the incomplete sequence lacking center (like 24 and 79). | |
| 7 | Single Wait | 1 | A hand completion solely waiting on a tile to form a pair. | |
| 8 | Self Draw | 1 | A hand completion by draw. | |
| 9 | Flower Tile | 1 | Whenever a player draws a flower or season tile, this is counted as 1 point. | |
| 10 | Ready Hand | 2 | A hand that is one tile away from winning. | |

| 11 | Dragon Pung | 2 | A hand with a triplet or quad of dragon tile. | |
| 12 | Concealed Hand | 2 | A hand without exposed melds and completed by seizing the discarded tile. | |
| 13 | All Chows | 2 | A hand with four sequences and a pair in characters. | |
| 14 | Tile Hog | 2 | A hand with four same tiles, other than a melded quad. | |
| 15 | Two Concealed Pungs | 2 | A hand with two concealed triplets or quads. | |
| 16 | Concealed Kong | 2 | A hand with a concealed quad. | |
| 17 | All Simples | 2 | A hand consisting of no character 1, 9 and honor tile. | |
| 18 | Outside Hand | 4 | A hand that includes terminals and honors in each meld, including the pair. | |
| 19 | Fully Concealed Hand | 4 | A hand without exposed melds and completed by drawing a tile. | Self Draw |
| 20 | Two Melded Kongs | 4 | A hand with two exposed quads. | Melded Kong |
| 21 | Last Tile | 4 | Winning on a tile that is the last of its kind. Three of the other tiles on the table already revealed to all players. | |
| 22 | Little Three Winds | 6 | A hand with two wind triplets or quads and a wind pair. | |
| 23 | All Pungs | 6 | A hand with four triplets or quads. | |
| 24 | Half Flush | 6 | A hand consisting of character and honor tiles. | |
| 25 | Two Dragon Pungs | 6 | A hand containing two dragon triplets or quads. | Dragon Pung |
| 26 | Two Concealed Kongs | 6 | A hand containing two concealed quads. | Concealed Kong |
| 27 | Melded Hand | 6 | Every set in the hand must be completed with tiles discarded by other players. | Single Wait |
| 28 | Out with Replacement Tile | 8 | Hand completion with supplemental tile when you melding quad. | Self Draw |
| 29 | Rob Kong | 8 | Winning off the tile that opponent adds to a melded triplet(to create a Kong). | Last Tile |
| 30 | Last Tile Claim | 8 | Winning off another player on the last tile (of the game). | |
| 31 | Pure Straight | 16 | A hand with three sequences 123, 456 and 789 in characters. | Short Straight, Two Terminal Chows |
| 32 | Pure Shifted Chows | 16 | Three sequences in characters each shifted either one or two numbers up from the last, but not a combination of both. | |
| 33 | All Flowers | 16 | A player has or draws all flower tiles. | Flower Tiles |
| 34 | Full Flush | 16 | A hand consisting of only characters. | |
| 35 | Three Concealed Pungs | 16 | A hand with three concealed quads or triplets. | Two Concealed Pungs |
| 36 | Four Honour Pungs | 24 | A hand with four honor triplets or quads. | All Pungs |
| 37 | Big Three Winds | 24 | A hand with three winds be triplets or quads. | Little Three Winds |

| 38 | Seven Pairs | 24 | A hand with seven pairs. | Concealed Hand, Single Wait, Fully Concealed Hand. |
|---|---|---|---|---|
| 39 | Pure Triple Chow | 24 | A hand with three identical sequences. | Pure Double Chow |
| 40 | Pure Shifted Pungs | 24 | A hand with three number triplets or quads with successive numbers. | Pure Triple Chow |
| 41 | Four Pure Shifted Chows | 32 | Four shifted number sequences, each shifted either one or two numbers up from the last, but not a combination of both. | Pure Shifted Chows, Short Straight, Two Terminal Chows, Pure Double Chow |
| 42 | Three Kongs | 32 | A hand with three quad melds. | Two Melded Kongs, Melded Kong, Two Concealed Kongs, Concealed Kong |
| 43 | All Terminals and Honours | 32 | A hand consisting of only terminal and honor tiles. | All Pungs, Outside Hand |
| 44 | Heavenly Ready Hand | 32 | A hand that is one tile away from winning at the beginning of the game. | Ready Hand |
| 45 | Quadruple Chow | 48 | A hand with four identical sequences. | Pure Triple Chow, Tile Hog, Pure Double Chow, Two Terminal Chow, Pure Shifted Pungs |
| 46 | Four Pure Shifted Pungs | 48 | A hand with four character triplets or quads with successive numbers. | Pure Shifted Pungs, Pure Triple Chow, All Pungs |
| 47 | Four Winds Seven Pairs | 48 | Seven pairs hand with four different wind pairs. | Seven Pairs, Concealed Hand, Single Wait, Fully Concealed Hand |
| 48 | Three Dragons Seven Pairs | 48 | Seven pairs hand with three different dragon pairs. | Seven Pairs, Concealed Hand, Single Wait, Fully Concealed Hand |
| 49 | Little Four Winds | 64 | A hand with three winds be triplets or quads, and the last wind be pair. | Big Three Winds, Little Three Winds |

| | | | | |
|---|---|---|---|---|
| 50 | Little Three Dragons | 64 | A hand with two dragons be triplets or quads, and the last dragon be pair. | Two Dragon Pungs, Dragon Pung |
| 51 | All Honours | 64 | A hand consisting of only honor tiles. | All Terminals And Honours, All Pungs, Four Honour Pungs, Outside Hand |
| 52 | Four Concealed Pungs | 64 | A hand with four concealed triplets or quads. | Concealed Hand, All Pungs, Three Concealed Pungs, Two Concealed Pungs, Fully Concealed Hand |
| 53 | Pure Terminal Chows | 64 | A hand with two Two Terminal Chows and a pair of number 5 in character. | All Chows, Seven Pairs, Full Flush, Pure Double Chow, Two Terminal Chows |
| 54 | Big Four Winds | 88 | A hand with triplets or quads of all four winds and an pair. | All Pungs, Little Three Winds, Big Three Winds, Four Honor Pungs |
| 55 | Big Three Dragons | 88 | A hand with triplets or quads of all three dragons. | Dragon Pung, Two Dragon Pungs |
| 56 | Nine Gates | 88 | Collecting number tiles 1112345678999 without melding, and completing with any tile of characters. | Full Flush, Concealed Hand, Fully Concealed Hand |
| 57 | Four Kongs | 88 | A hand with four quad melds. | Three Kongs, Two Melded Kongs, Melded Kong, Single Wait, Concealed Kong, Two Concealed Kongs, All Pungs |
| 58 | Seven Shifted Pairs | 88 | Seven pairs hand with successive seven numbers in characters. | Seven Pairs, Single Wait, Concealed Hand, Full Flush, Fully Concealed Hand |

| 59 | Upper Four | 88 | A hand consisting of character tiles of 6, 7, 8 or 9 | Seven Pairs |
|----|------------|-----|---------------------------------------------------------|-------------|
| 60 | Lower Four | 88 | A hand consisting of character tiles of 1, 2, 3 or 4 | Seven Pairs |
| 61 | Big Seven Honours | 88 | Seven pairs hand with four different wind pairs and three different dragon pairs. | Seven Pairs, Four Winds Seven Pairs, Three Dragons Seven Pairs, Outside Hand, Single Wait, Concealed Hand, Fully Concealed Hand, All Honours |
| 62 | Heavenly Hand | 88 | The dealer draws a winning hand at the beginning of the game. | Self Draw, Concealed Hand |
| 63 | Earthly Hand | 88 | A player completes a winning hand with the dealer's first discard and in most variants, provided the dealer does not draw a quad. | Self Draw, Concealed Hand |
| 64 | Humanly Hand | 88 | A player completes a winning hand with the opponent player's first discard. And before that any action of Chow, Pong or Kong is not available. | |

Table 3: The categories of legal hands in ascending order of corresponding points. The winning points of a legal hand are obtained by summing the points of the matched categories in reverse order while excluding the conflict categories.

## A.2 THE STATE AND ACTION SPACE

The state space size of 1-on-1 Mahjong, shown as the infoset count in Figure 4, is approximately $10^{74}$. Yet, the state space of 1-on-1 Mahjong is not as easily abstracted as in poker. The primary reason is that a single tile difference in the state could significantly impact the policy, e.g., making a legal hand illegal and vice versa. In contrast, states that have similar strength in poker could share a common policy. For instance, the optimal preflop policy could be very similar for "Ace-Four" and "Ace-Three" in poker. Another reason is that a state in 1-on-1 Mahjong is divided into different information groups, as demonstrated in Figure 7(a). Different information groups have significantly different meanings. For instance, one group denotes the player's hand, which is invisible to the opponent, while another one denotes the player's discarded tiles, which are visible to both players.

There are 105 different actions in total, as demonstrated in Table 2. The number of legal actions in a state is relatively small compared to poker. Yet, the game length in 1-on-1 Mahjong is larger than that in poker. Players can decide up to about 40 sequential actions in 1-on-1 Mahjong, whereas most 1-on-1 poker games end within 10 steps. As a result, the reaching probability of states in 1-on-1 Mahjong may vary more significantly than that in poker.

## A.3 THE EFFECTS OF A LARGER INFOSET SIZE AND A LONGER GAME LENGTH

As shown in Figure 4, 1-on-1 Mahjong has a larger infoset size than poker. The infoset size does not seem to have an influence on the convergence of a tabular CFR (Zinkevich et al., 2008). However, when trajectory sampling and function approximation are used together, the situation may be different. To be more specific, in a trajectory sampling algorithm, the variance of the sampled instantaneous counterfactual value (regret) of a larger infoset may tend to be higher, which may have a large

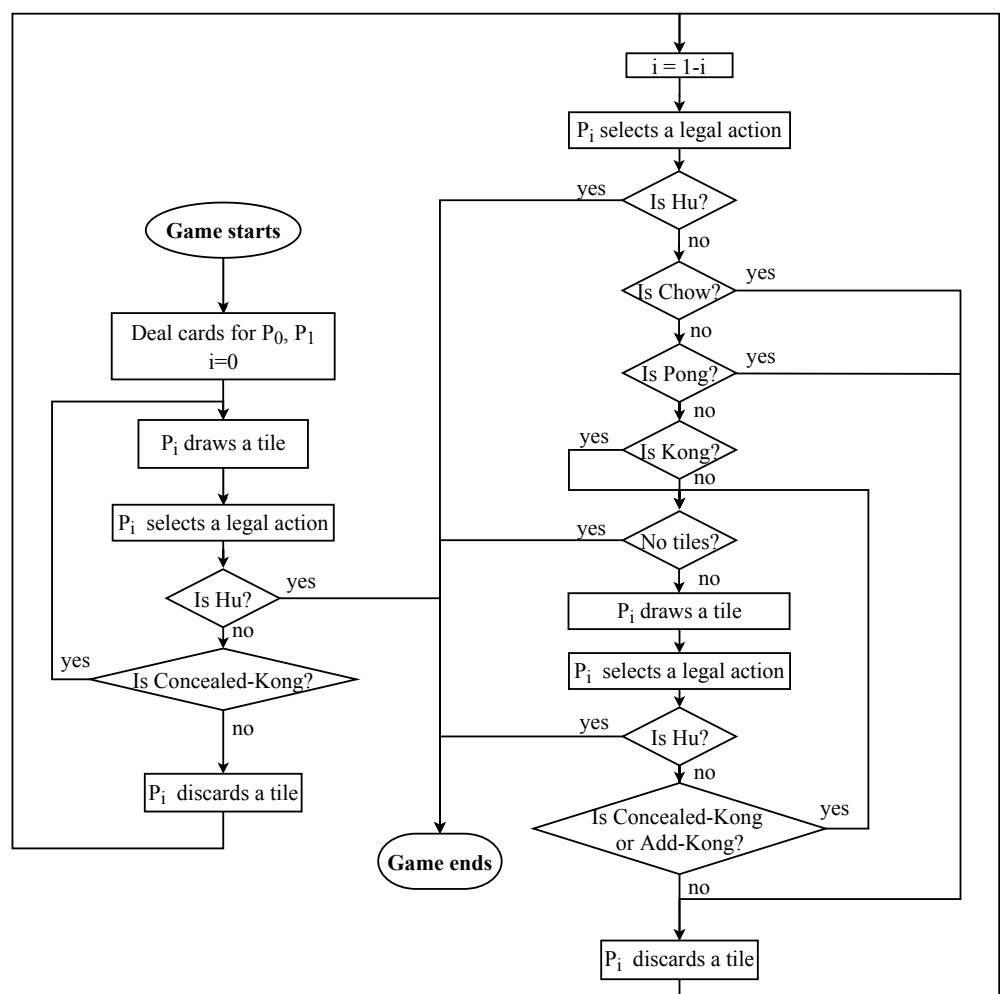

Figure 6: A flow chart of the 1-on-1 Mahjong game.

influence on performance when neural network function approximation is used. In other words, 1-on-1 Mahjong may be complementary to poker in evaluating algorithms using deep neural networks and only trajectory samples.

The game length has a direct impact on the sampling methods used. For poker, which has a relatively short game length, methods that sample multiple actions in a state are very common in the literature. Yet, sampling multiple actions consistently in game trees with long episodes is certainly prohibitive, as the number of samples goes exponentially with the game length. 1-on-1 Mahjong has a maximal game length about $40$, which may be daunting for methods that try to sample multiple actions in a state. In other words, 1-on-1 Mahjong may be complementary to poker in evaluating algorithms when only trajectory samples are allowed.

## B   THE MODEL DESIGN OF OUR 1-ON-1 MAHJONG AGENT JUEJONG

The model of JueJong is an end-to-end neural network that takes all relevant information as input and outputs both the probabilities of all actions and the state value. This is different from five separated neural networks in Suphx (Li et al., 2020b), representing five different types of actions. Also, we train JueJong from zero by pure self-play using ACH, while the five neural networks in Suphx were trained by supervised learning on human data, with only the "Discard" network further enhanced by RL. Figure 7 gives an overview of the model design of JueJong.

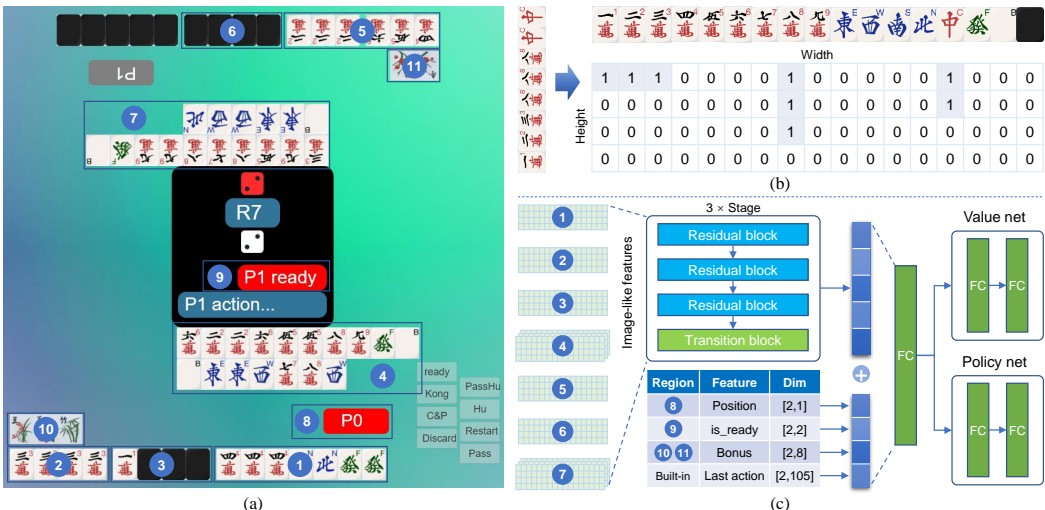

Figure 7: **(a)** The graphical user interface of 1-on-1 Mahjong with marked regions representing different groups of information, which are encoded in either image-like features or one-hot features. **(b)** The image-like feature encoding scheme. **(c)** The model architecture design.

The marked regions in Figure 7(a) summarize the information an agent can observe. Regions 1 to 7 are encoded in image-like features, which represent the player's hand, the player's Chow, Pong, and Kong, the player's concealed-Kong, the player's Discard, the opponent's Chow, Pong, and Kong, the opponent's concealed-Kong, and the opponent's Discard respectively. A feature map, as shown in Figure 7(b), of height 4 and width 17 is employed, where a 0 or 1 in the $i$th row and $j$th column means whether there are $i$ tiles of the $j$th type in the input set of tiles. Therefore, a single or two successive convolutional layers with $3 \times 3$ kernels efficiently derive row and column combinations. Note that, in Suphx (Li et al., 2020b), the feature maps are one-dimensional vectors, where $1 \times 3$ kernels are used. The black tile in Figure 7(b) is designed exclusively for Region 6 to indicate how many concealed-Kongs the opponent has. For discarded tiles in Region 4 or 7, 24 such feature maps are used to indicate the latest 24 discarded tiles in order, with one feature map encoding one discard tile. All the feature maps are concatenated in the channel dimension, and therefore the final image-like features are in the shape of $4 \times 17 \times 53$ ($h \times w \times c$).

Regions 8 to 11 are encoded in one-hot features, which represent the player's position (0 or 1), the $is\_ready$ state of both players, and the bonus tiles of both players. Additionally, the last action (not shown as a region in Figure 7(a)) for each player is encoded in a one-hot feature vector as well.

We apply residual blocks (He et al., 2016) to transform the image-like features. There are totally 3 stages with 3 residual blocks and 1 transition layer in each stage, as shown in Figure 7(c). Each block contains two convolutional layers with kernel size $3 \times 3$. The transition layers are point-wise convolutions that scale the number of output channels to 64, 128, and 32, respectively for each stage. Subsequently, the transformed image-like features are reshaped to a vector, which is concatenated with the one-hot feature vectors. A fully-connected layer (dimension 1024) is then used to transform the concatenated feature vector, and two branches with two fully-connected layers (dimension $512 \times 512$) each output the action probabilities and the state value respectively. Besides, we apply batch normalization (Ioffe & Szegedy, 2015) and ReLU non-linearity after all convolutional layers and fully-connected layers.

## C THEORETICAL PROPERTIES OF NW-CFR

### C.1 PROOF FOR THEOREM 1

In order to prove Theorem 1, we first prove the following Lemma.

**Lemma 1.** *For a weighted CFR, if $w_t(s) \in [w_l(s), w_h(s)], 0 < w_l(s) \leq w_h(s) \leq 1$, then*

$$\max_{a \in \mathcal{A}(s)} \sum_{k=1}^{t} r_k^c(s, a) \leq \frac{\max_{a \in \mathcal{A}(s)} R_t^w(s, a)}{w_h(s)} + \frac{(w_h(s) - w_l(s))|\mathcal{A}(s)|\Delta t}{w_h(s)}. \qquad (4)$$

*Proof.* First, for any state $s \in S$ and action $a \in \mathcal{A}(s)$, we have

$$\begin{aligned}
R_t^w(s, a) &:= \sum_{k=1}^{t} w_t(s) r_k^c(s, a) \\
&= \sum_{k:r_k^c(s,a) \geq 0} w_k(s)|r_k^c(s, a)| - \sum_{k':r_{k'}^c(s,a) < 0} w_{k'}(s)|r_{k'}^c(s, a)| \qquad (5) \\
&\geq \sum_{k:r_k^c(s,a) \geq 0} w_l(s)|r_k^c(s, a)| - \sum_{k':r_{k'}^c(s,a) < 0} w_h(s)|r_{k'}^c(s, a)|.
\end{aligned}$$

So,

$$R_t^w(s, a) \geq w_h(s) \sum_{k=1}^{t} r_k^c(s, a) - (w_h(s) - w_l(s)) \sum_{k':r_{k'}^c(s,a) \geq 0} |r_{k'}^c(s, a)|. \qquad (6)$$

In other words,

$$\sum_{k=1}^{t} r_k^c(s, a) \leq \frac{R_t^w(s, a)}{w_h(s)} + \frac{w_h(s) - w_l(s)}{w_h(s)} \sum_{k':r_{k'}^c(s,a) \geq 0} |r_{k'}^c(s, a)| \qquad (7)$$

So,

$$\max_{a \in \mathcal{A}(s)} \sum_{k=1}^{t} r_k^c(s, a) \leq \frac{\max_{a \in \mathcal{A}(s)} R_t^w(s, a)}{w_h(s)} + \frac{(w_h(s) - w_l(s))\Delta t}{w_h(s)}. \qquad (8)$$

$\square$

Then we can prove the Theorem:

*Proof.* For NW-CFR, the policy at iteration $t$ is generated according to Hedge:

$$\pi_t(s, a) = \frac{e^{\eta(s) R_{t-1}^a(s,a)}}{\sum_{a'} e^{\eta(s) R_{t-1}^a(s,a')}}, \qquad (9)$$

where

$$R_{t-1}^a(s, a) = \sum_{k=1}^{t-1} f_p^{\mu_k}(s) r_k^c(s, a). \qquad (10)$$

Meanwhile, weighted CFR with Hedge generates policy at iteration $t$ according to

$$\pi_t(s, a) = \frac{e^{\eta(s) R_{t-1}^w(s,a)}}{\sum_{a'} e^{\eta(s) R_{t-1}^w(s,a')}}, \qquad (11)$$

where

$$R_{t-1}^w(s, a) = \sum_{k=1}^{t-1} w_k(s) r_k^c(s, a). \qquad (12)$$

So, when $w_k(s) = f_p^{\mu_k}(s) \geq w_l(s) > 0$ for any $s \in \mathcal{S}$ and $k > 0$, NW-CFR and weighted CFR with Hedge generate the same policy at each iteration, and therefore NW-CFR is equivalent to weighted CFR with Hedge.

Then, we can prove the convergence property of NW-CFR. Let

$$S_t^w(s) = \sum_{a \in \mathcal{A}(s)} e^{\eta(s) R_t^w(s,a)}. \qquad (13)$$

We have

$$
\begin{aligned}
\ln \frac{S_t^w}{S_0^w} &= \ln \sum_{a \in \mathcal{A}(s)} e^{\eta(s)R_t^w(s,a)} - \ln|\mathcal{A}(s)| \\
&\geq \ln \left( \max_{a \in \mathcal{A}(s)} e^{\eta(s)R_t^w(s,a)} \right) - \ln|\mathcal{A}(s)| \\
&= \eta(s) \max_{a \in \mathcal{A}(s)} R_t^w(s,a) - \ln|\mathcal{A}(s)|.
\end{aligned}
\tag{14}
$$

Meanwhile, for each $k = 1, \ldots, t$,

$$
\begin{aligned}
\ln \frac{S_k^w}{S_{k-1}^w} &= \ln \left( \frac{\sum_{a \in \mathcal{A}(s)} e^{\eta(s)R_{k-1}^w(s,a)} e^{\eta(s)w_k(s)r_k^c(s,a)}}{\sum_{a \in \mathcal{A}(s)} e^{\eta(s)R_{k-1}^w(s,a)}} \right) \\
&= \ln \left( \sum_{a \in \mathcal{A}(s)} \pi_k(s,a) e^{\eta(s)w_k(s)r_k^c(s,a)} \right).
\end{aligned}
\tag{15}
$$

Since

$$
\ln \mathbb{E}[e^{sX}] \leq s\mathbb{E}X + \frac{s^2(b-a)^2}{8},
\tag{16}
$$

$$
\ln \frac{S_k^w}{S_{k-1}^w} \leq \eta(s) \sum_{a \in \mathcal{A}(s)} \pi_k(s,a)w_k(s)r_k^c(s,a) + \frac{\eta^2(s)\Delta^2(s)w_k^2(s)}{8}.
\tag{17}
$$

Note that $r_k^c(s,a) = v_k^c(s,a) - \sum_{a \in \mathcal{A}(s)} \pi_k(s,a)v_k^c(s,a)$. So

$$
\sum_{a \in \mathcal{A}(s)} \pi_k(s,a)w_k(s)r_k^c(s,a) = 0.
\tag{18}
$$

Therefore,

$$
\ln \frac{S_k^w}{S_0^w} \leq \frac{\eta^2(s)\Delta^2(s)\sum_{k=1}^t w_k^2(s)}{8}.
\tag{19}
$$

So,

$$
\eta(s) \max_{a \in \mathcal{A}(s)} R_t^w(s,a) - \ln|\mathcal{A}(s)| \leq \frac{\eta^2(s)\Delta^2(s)\sum_{k=1}^t w_k^2(s)}{8},
\tag{20}
$$

i.e.,

$$
\max_{a \in \mathcal{A}(s)} R_t^w(s,a) \leq \frac{\ln|\mathcal{A}(s)|}{\eta(s)} + \frac{\eta(s)\Delta^2(s)\sum_{k=1}^t w_k^2(s)}{8}.
\tag{21}
$$

According to Lemma 1,

$$
\max_{a \in \mathcal{A}(s)} R_t^c(s,a) \leq \frac{\ln|\mathcal{A}(s)|}{\eta(s)w_h(s)} + \frac{\eta(s)\Delta^2(s)\sum_{k=1}^t w_k^2(s)}{8w_h(s)} + \frac{(w_h(s) - w_l(s))\Delta t}{w_h(s)}.
\tag{22}
$$

When $\eta(s) = \sqrt{8\ln|\mathcal{A}(s)|/\{[w_h(s)]^2\Delta^2(s)T\}}$ and $w_t(s) \in [w_l(s), w_h(s)] \subset (0,1], t = 1, \ldots, T$, we have

$$
\begin{aligned}
\max_{a \in \mathcal{A}(s)} R_T^c(s,a) &\leq \sqrt{\frac{\ln|\mathcal{A}(s)|\Delta^2(s)w_h^2(s)T}{2w_h^2(s)}} + \frac{(w_h(s) - w_l(s))\Delta T}{w_h(s)} \\
&\leq \Delta\sqrt{\frac{T}{2}\ln|\mathcal{A}(s)|} + \frac{(w_h(s) - w_l(s))\Delta T}{w_h(s)}.
\end{aligned}
\tag{23}
$$

According to Theorem 2 in Zinkevich et al. (2008), the total regret

$$
\begin{aligned}
R_T &\leq \sum_{s \in S} \max_{a \in \mathcal{A}(s)} [R_T^c(s,a)]^+ \\
&\leq \sum_{s \in S} \left( \Delta\sqrt{\frac{T}{2}\ln|\mathcal{A}(s)|} + \frac{(w_h(s) - w_l(s))\Delta T}{w_h(s)} \right) \\
&\leq |\mathcal{S}|\Delta\sqrt{\frac{T}{2}\ln|\mathcal{A}|} + \Delta T \sum_{s \in \mathcal{S}} \frac{w_h(s) - w_l(s)}{w_h(s)}.
\end{aligned}
\tag{24}
$$

As a result, according to the folk theorem in Zinkevich et al. (2008), the average policy has $\epsilon$ exploitability, where

$$\epsilon = \frac{1}{|\mathcal{P}|}\sum_{p\in\mathcal{P}}\frac{R_{p,T}}{T} \leq |\mathcal{S}|\Delta\sqrt{\frac{1}{2T}\ln|\mathcal{A}|} + \Delta\sum_{s\in\mathcal{S}}\frac{w_h(s)-w_l(s)}{w_h(s)}. \tag{25}$$

$\square$

### C.2 PROOF FOR COROLLARY 1

*Proof.* When the behavioral policy $\mu_{p,k}$ of each player $p \in \mathcal{P}$ is constant across iterations $\forall k > 0$, the reaching probability $f_p^{\mu_k}(s)$ of any state $s \in \mathcal{S}$ is also constant. Assume $f_p^{\mu_k}(s) = w(s)$, then,

$$R_{t-1}^a(s,a) = \sum_{k=1}^{t-1} f_p^{\mu_k}(s) r_t^c(s,a) = w(s)\sum_{k=1}^{t-1} r_t^c(s,a) = w(s)R_{t-1}^c(s,a). \tag{26}$$

In other words, $R_{t-1}^a(s,a)$ is equal to the cumulative counterfactual regret scaled by a time-invariant weight $w(s)$. Hence, the policy at iteration $t$ is

$$\pi_t(a|s) = \frac{e^{\eta(s)w(s)R_t^c(s,a)}}{\sum_{a'} e^{\eta(s)w(s)R_{t-1}^c(s,a')}} = \frac{e^{\eta'(s)R_t^c(s,a)}}{\sum_{a'} e^{\eta'(s)R_{t-1}^c(s,a')}}, \tag{27}$$

where the new $\eta'(s)$ is set to $\sqrt{8\ln|\mathcal{A}(s)|/[\Delta^2(s)T]}$. As a result, for constant $\mu_{p,k}$, NW-CFR is equivalent to CFR with Hedge when $y(a|s;\theta_t)$ is sufficiently close to $R_t^a(s,a)$. Furthermore, since $w_l(s) = f_p^{\mu_k}(s) = w_h(s)$, the second term in Equation 25 vanishes, i.e.,

$$\epsilon \leq |\mathcal{S}|\Delta\sqrt{\frac{1}{2T}\ln|\mathcal{A}|}. \tag{28}$$

As a result, the exploitability bound of CFR with Hedge is recovered. $\square$

## D EXPERIMENTAL RESULTS OF THE WEIGHTED CFR

As stated in Section 5, ACH is a practical implementation of NW-CFR, which is a straightforward neural extension to the weighted CFR defined in Definition 1, together with $w_t(s) = f_p^{\mu_t}(s)$ and Hedge. In order to investigate the behavior of weighted CFR, in this section, we instantiate multiple weighted CFR algorithms with different settings of the weight $w_t(s)$ by varying $\mu_{p,t}$, since $f_p^{\mu_t}(s)$ depends only on $\mu_{p,t}$. Note that the weighted CFR traverses the full game tree at every iteration and that $\mu_{p,t}$ is only used to calculate the state reaching probability $f_p^{\mu_t}(s)$. We test these algorithms on three small IIGs in OpenSpiel: Kuhn poker, Leduc poker, and Liar's Dice.

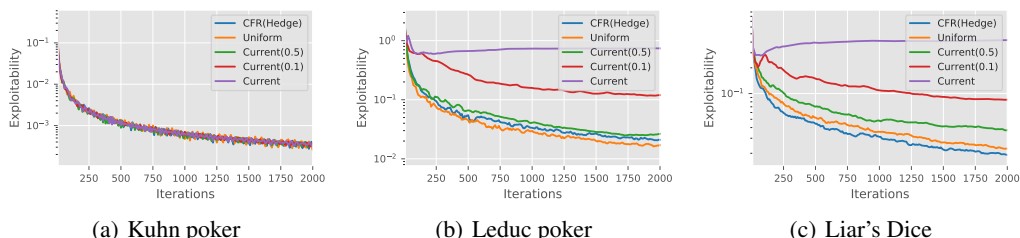

|     (a) Kuhn poker     |     (b) Leduc poker     |     (c) Liar's Dice     |

Figure 8: Exploitability of the weighted CFR with $w_t(s) = f_p^{\mu_t}(s)$ and CFR (i.e., the weighted CFR with $w_t(s) = 1.0$). The probability $f_p^{\mu_t}(s)$ is determined by the behavior policy $\mu_{p,t}$. The setting of $\mu_{p,t}$ for each line is given in the legend, in which "Uniform" means $\mu_{p,t}(s) =$ the uniform policy; "Current" means $\mu_{p,t}(s) = \pi_{p,t}(s)$; "Current$(x)$" means $\mu_{p,t}(s) = x$Uniform $+ (1-x)\pi_{p,t}(s)$. Note that the exploitability is reported with regard to the average policy.

As shown in Figure 8, the weighted CFR with $w_t(s)$ induced by a stationary $\mu_{p,t}$ (i.e., the uniform policy) converges at the same pace with CFR(Hedge). As a result, Corollary 1 is verified on the three

small benchmarks. Also, as Theorem 1 states, the exploitability of the weighted CFR is influenced by the range of $w_t(s) = f_p^{\mu_t}(s)$. This is experimentally demonstrated in Figure 8 by setting $\mu_{p,t}$ to a mixed policy between the current policy $\pi_{p,t}$ and the uniform policy. We can see that the weighted CFR still performs competitively with CFR(Hedge), when $\mu_{p,t}(s) = 0.5\text{Uniform} + 0.5\pi_{p,t}(s)$.

## E    ACH: A PRACTICAL IMPLEMENTATION OF NW-CFR

To address the practical issues mentioned in Section 5, we provide a practical and parallel implementation of NW-CFR, i.e., ACH, which employs a framework of decoupled acting and learning, similar to IMPALA (Espeholt et al., 2018). ACH maintains a policy net $y(a|s; \theta)$ and a value net $V(s; \omega)$, where $\theta$ and $\omega$ share a large portion of parameters (see Figure 7). Both players use the same $\theta$ and $\omega$. We do not use an additional time-invariant behavioral policy for sampling actions. Instead, we use the current policy $\pi_t$, i.e., $\mu_{p,t} = \pi_{p,t}, \forall p \in \mathcal{P}$. As a result, we can use the same samples to train both the value net and the policy net. Also, $\eta(s)$ is incorporated into the learned target value in the policy net, so the policy $\pi(a|s)$ is obtained by directly softmaxing on $y(a|s; \theta)$.

The advantage $A(s, a)$ is estimated by GAE($\lambda$) (Schulman et al., 2016), using sampled rewards and $V(s; \omega)$, for only sampled states and actions. The value and policy nets are updated as soon as a mini-batch of samples is available. In other words, we update $\theta$ and $\omega$ once using a single mini-batch at each iteration. As a result, the policy loss reduces to $\mathcal{L}_\pi(s) = \eta(s)\frac{y(a|s;\theta)}{\pi_{old}(a|s)}A(a, s)$, where $\frac{1}{\pi_{old}(a|s)}$ accounts for the fact that the action $a$ was sampled using $\pi_{old}(a|s)$. ACH handles asynchronous training with the importance ratio clipping $[1 - \varepsilon, 1 + \varepsilon]$ of PPO (Schulman et al., 2017). To avoid numerical issues, the mean $\bar{y}(\cdot|s; \theta)$ is subtracted from the policy output, which is then clipped within a range $[-l^{th}, l^{th}]$. The pseudocode of ACH is given in Algorithm 2.

---

**Algorithm 2: ACH**

---

Initialize the policy and critic parameters: $\theta$ and $\omega$.
Start multiple actor and learner threads in parallel.
**Actors**:
**while** *true* **do**
> Fetch the latest model from the learners.
> Generate samples via self-play in the form: $[a, s, A(s, a), G, \pi_{old}(a|s)]$.
> Send the samples to the replay buffer.

**Learners**:
**for** $t \in 1, 2, 3, ...$ **do**
> Fetch a mini-batch of samples from the replay buffer.
> $\mathcal{L}_{sum} = 0$.
> **for** *each sample* $[a, s, A(s, a), G, \pi_{old}(a|s)] \in$ *the mini-batch* **do**
> > $c = \begin{cases} \mathbb{1}\{\frac{\pi(a|s;\theta)}{\pi_{old}(a|s)} < 1 + \varepsilon\}\mathbb{1}\{y(a|s;\theta) - \bar{y}(\cdot|s;\theta) < l^{th}\} & \text{if } A(s,a) \geq 0, \\ \mathbb{1}\{\frac{\pi(a|s;\theta)}{\pi_{old}(a|s)} > 1 - \varepsilon\}\mathbb{1}\{y(a|s;\theta) - \bar{y}(\cdot|s;\theta) > -l^{th}\} & \text{if } A(s,a) < 0. \end{cases}$
> > $\mathcal{L}_{sum} \mathrel{+}= -c\eta(s)\frac{y(a|s;\theta)}{\pi_{old}(a|s)}A(a, s) + \frac{\alpha}{2}[V(s;\omega) - G]^2 + \beta\sum_a \pi(a|s;\theta)\log\pi(a|s;\theta)$.
> Update $\theta$ and $\omega$ once using gradient on $\mathcal{L}_{sum}$.

---

We employ an entropy loss to encourage exploration during training and hopefully the convergence of current policy to a NE (Srinivasan et al., 2018). ACH updates $\theta$ and $\omega$ simultaneously, and the overall loss is:

$$\mathcal{L}_{ACH} = -c\eta(s)\frac{y(a|s;\theta)}{\pi_{old}(a|s)}A(a, s) + \frac{\alpha}{2}[V(s;\omega) - G]^2 + \beta\sum_a \pi(a|s;\theta)\log\pi(a|s;\theta). \tag{29}$$

Theoretically, $y(a|s'; \theta)$ for non-sampled states $s'$ should also be trained with the target $y(a|s'; \theta_{old})$. Yet, in ACH, we only update $\theta$ once using a single mini-batch at each iteration. Therefore, $\theta$ is equal to $\theta_{old}$ before the update. In other words, the policy loss for non-sampled states is 0 in ACH.

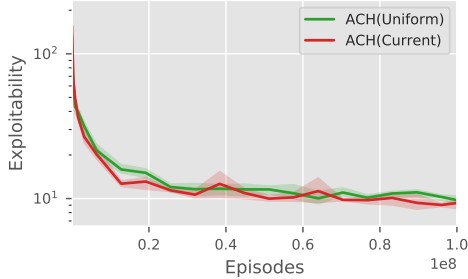

Figure 9: The exploitability of the current policy in ACH with different behavior policies on FHP. We report the mean as solid curves and the range as shaded regions across 3 independent runs.

## F    THE EFFECT OF THE BEHAVIOR POLICY ON ACH IN FHP

As we noted in the paper, the behavior policy $\mu_{p,t}$ in ACH could be set to either the current policy $\pi_{p,t}$ or simply a uniform sampling policy. As Corollary 1 states, a tighter bound on the exploitability of the average policy of NW-CFR can be obtained, if $\mu_{p,t}$ is stationary over iterations. However, since we have decided to use the current policy (trained with an entropy regularization) for evaluation in ACH, the effect of the behavior policy is unclear from a theoretical perspective. Hence, we conduct an experiment to investigate this using FHP, which is a non-trivial poker benchmark but still has the property that the exact exploitability of an agent can be efficiently computed.

In Figure 9, we compare the exploitability of the current policy of ACH on FHP, by setting the behavior policy in ACH to either the current policy or a uniform sampling policy. From the comparison, it seems that the performance of the current policy of ACH is not sensitive to the choice of behavior policy. One reason might be that the additional entropy loss forces the current policy to be stable and prone to a uniform random policy. Another reason might be that the current policy instead of the average policy is evaluated. We will investigate this in more depth in the future.

## G    ADDITIONAL RESULTS ON SMALL IIG BENCHMARKS IN OPENSPIEL

We further evaluate ACH and compare it with A2C, RPG, and NeuRD on three benchmarks from OpenSpiel: Kuhn poker, Leduc poker, and Liar's Dice. All the experiments were run single-threaded on a 2.24GHz CPU. We use the network architecture provided in OpenSpiel, which has a 128-neurons fully-connected layer followed by ReLU and two separate linear layers for the policy and the state/action value. For A2C, RPG, and NeuRD, we use the default hyper-parameters in OpenSpiel. ACH shares most of the hyper-parameters with A2C. All the hyper-parameters are listed in the Appendix H.3.

The exploitability of an agent is exactly calculated using tools in OpenSpiel. For each method, we compute the exploitability of each agent every $1e5$ training steps, the results of which are plotted in Figure 10. Clearly, ACH converges significantly faster and achieves a lower exploitability than other methods across the three benchmarks. There is still some gap between $0$ and the exploitability ACH converges to. This may due to the neural network approximation error and the fact that we use the current policy instead of the average policy for the evaluation. As expected, A2C has the worst performance, since it is designed for single-agent environments. Moreover, the superiority of ACH is most significant on the Liar's Dice benchmark, which is the most complex one of the three benchmarks.

As a complement, we also present the head-to-head performance of A2C, RPG, NeuRD, and ACH on the three benchmarks in OpenSpiel. As demonstrated in Table 4, the agent of ACH won all other agents across the three benchmarks.

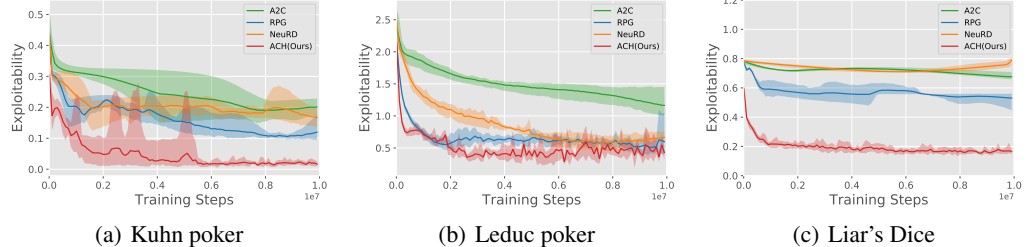

(a) Kuhn poker        (b) Leduc poker        (c) Liar's Dice

Figure 10: The training curves of each agent on the three benchmarks from OpenSpiel. We report the mean as solid curves and the range of the exploitability across 8 independent runs as shaded regions. We notice that there exists some discrepancy between the NeuRD results on Kuhn poker and Leduc poker reported here and those reported in Hennes et al. (2020) and Lanctot et al. (2019). The reason might be due to NeuRD's sensitivity to running environments including hyper-parameters and random seeds.

| | Kuhn poker | | | Leduc poker | | | Liars Dice | | |
|---|---|---|---|---|---|---|---|---|---|
| | **A2C** | **RPG** | **NeuRD** | **A2C** | **RPG** | **NeuRD** | **A2C** | **RPG** | **NeuRD** |
| **RPG** | **-0.024**±0.20 | - | - | **0.521**±0.23 | - | - | **0.192**±0.14 | - | - |
| **NeuRD** | **-0.096**±0.16 | **-0.087**±0.16 | - | **0.569**±0.16 | **-0.108**±0.17 | - | **-0.226**±0.05 | **-0.355**±0.12 | - |
| **ACH** | **0.104**±0.05 | **0.115**±0.05 | **0.118**±0.05 | **0.495**±0.22 | **0.050**±0.14 | **0.117**±0.15 | **0.275**±0.05 | **0.122**±0.05 | **0.407**±0.06 |

Table 4: Mean (± standard deviation) of the average winning scores of the row agents against the column agents. The mean and the standard deviation are estimated by 8 independent runs. In each run, the average winning scores are obtained via $10,000$ head-to-head plays. All the agents are selected at the $1e7$th training step.

## H HYPER-PARAMETERS

### H.1 HYPER-PARAMETERS FOR THE 1-ON-1 MAHJONG EXPERIMENT

We used the Adam optimizer (Kingma & Ba, 2014) for the experiments on the 1-on-1 Mahjong benchmark. We performed a mild hyper-parameter search on PPO and used the best setting for the shared hyper-parameters of all methods (PPO, RPG, NeuRD, and ACH). Table 5 gives an overview of hyper-parameters for each method. Also, we report the performance of the current policy, instead of the average policy, for each method.

| Parameter | Range | Best |
|---|---|---|
| *Shared* | | |
|     Ratio clip ($\varepsilon$) | - | 0.5 |
|     GAE ($\lambda$) | - | 0.95 |
|     Learning rate | {2.5e-3, 2.5e-4} | 2.5e-4 |
|     Discount factor ($\gamma$) | - | 0.995 |
|     Value loss coefficient ($\alpha$) | - | 0.5 |
|     Entropy coefficient ($\beta$) | {1e-1, 1e-2} | 1e-2 |
|     Batch size | {4096, 8192} | 8192 |
| *NeuRD* | | |
|     Logit threshold ($l^{th}$) | - | 6.0 |
| *ACH* | | |
|     Logit threshold ($l^{th}$) | - | 6.0 |
|     Hedge coefficient ($\eta(s)$) | {1.0, 1e-1} | 1.0 |

Table 5: The hyper-parameters used for the 1-on-1 Mahjong experiment.

## H.2 HYPER-PARAMETERS FOR THE FHP EXPERIMENT

We used the Adam optimizer (Kingma & Ba, 2014) for the experiments on FHP. All the experiments on FHP were run multi-threaded and synchronously using ten 2.24GHz CPUs. We update the neural networks of ACH and other methods (A2C, RPG, and NeuRD) every 1000 episodes, with a batch consisting all the samples collected within the latest 1000 episodes. Since the average game length of FHP is around 2, the batch size is roughly 2000. We performed a mild hyper-parameter search for ACH, A2C, RPG, and NeuRD, as shown in Table 6. Note that we do not need to clip the advantages in a synchronous method, so the ratio clip hyper-parameter $\varepsilon$ is not needed. We multiply the rewards in FHP with a reward normalizer, as the rewards in FHP are in the range $[-700, 700]$. Besides, we found that A2C, RPG, and NeuRD are very sensitive to the entropy coefficient, and we had to use a larger entropy coefficient in these algorithms than ACH. For OS-DCFR and DREAM, we use the same hyper-parameters presented in Steinberger et al. (2020). An overview of the hyper-parameters on FHP is given in Table 6. Note that we report the performance of the current policy for ACH, A2C, RPG, and NeuRD, while the average policy is used for evaluation in OS-DCFR and DREAM.

| Parameter | Range | Best |
|---|---|---|
| *Shared* | | |
| GAE ($\lambda$) | - | 0.95 |
| Learning rate | {1e-3, 1e-4} | 1e-4 |
| Discount factor ($\gamma$) | - | 0.995 |
| Value loss coefficient ($\alpha$) | - | 2.0 |
| Batch size | - | $\approx 2000$ |
| Entropy coefficient ($\beta$) | {1e-2, 3e-2, 5e-2} | 5e-2 |
| Reward normalizer | - | 0.002 |
| *NeuRD* | | |
| Logit threshold ($l^{th}$) | {2.0, 4.0} | 2.0 |
| *ACH* | | |
| Logit threshold ($l^{th}$) | {2.0, 4.0} | 2.0 |
| Entropy coefficient ($\beta$) | {1e-2, 3e-2, 5e-2} | 3e-2 |
| Hedge coefficient ($\eta(s)$) | {1.0, 0.1} | 1.0 |

Table 6: The hyper-parameters used for the FHP experiment.

## H.3 HYPER-PARAMETERS FOR THE EXPERIMENT ON OPENSPIEL

We used stochastic gradient descent with a constant learning rate for all the experiments on benchmarks from OpenSpiel. For A2C and RPG, we used the default implementations and hyper-parameters in OpenSpiel. NeuRD was originally implemented using the counterfactual regret in OpenSpiel. We re-implemented NeuRD using predicted advantages and set the shared hyper-parameters of NeuRD identical to those in RPG. All methods employ an entropy loss to encourage exploration during training and hopefully the convergence of the current policy to a NE. Also, we report the performance of the current policy, instead of the average policy, for each method.

| Parameter | Range | Best |
|---|---|---|
| Learning rate | {1e-3, 5e-3} | 1e-3 |
| Value loss coefficient ($\alpha$) | {1.0, 2.0} | 2.0 |
| Hedge coefficient ($\eta(s)$) | {1.0, 1e-1, 1e-2} | 1.0 |

Table 7: The hyper-parameter search ranges and best settings of ACH for the OpenSpiel experiment.

In the OpenSpiel implementations, the value parameters are updated separately and more frequently compared to the policy parameters for A2C, RPG, and NeuRD. Yet, the value loss and the policy loss are combined in ACH, and all parameters are updated simultaneously, as illustrated in Algorithm 2.

| Parameter | A2C | RPG | NeuRD | ACH |
|---|---|---|---|---|
| Batch size | 4 | 64 | 64 | 64 |
| Critic learning rate | 1e-4 | 1e-2 | 1e-2 | - |
| Policy learning rate | 1e-4 | 1e-2 | 1e-2 | - |
| # Critic updates per policy update | 32 | 32 | 32 | - |
| Entropy coefficient ($\beta$) | 1e-2 | 1e-2 | 1e-2 | 1e-2 |
| Logit threshold ($l^{th}$) | - | - | 2.0 | 2.0 |

Table 8: The hyper-parameters used for the OpenSpiel experiment.

Also note that, in a single-threaded training environment, the actor and the learner run in sequence. As a result, $\pi(a|s;\theta)$ is always identical to $\pi_{old}(a|s)$ in ACH in Algorithm 2.

We performed a mild hyper-parameter search for ACH, which is illustrated in Table 7. The final hyper-parameters used for each method are listed in Table 8, with 3 additional hyper-parameters of ACH listed in the "Best" column in Table 7.

## I    THE RELATIONSHIP BETWEEN ACH AND SUPHX

Recently, Suphx (Li et al., 2020b) has achieved stunning performance on Japanese Riichi Mahjong. The development of Suphx is enabled by a novel integration of existing supervised learning and RL methods in addition to some newly developed techniques. Three new techniques were introduced in Suphx: global reward prediction, oracle guiding, and run-time policy adaptation. The global reward prediction technique is to handle the multi-round game situation, which is irrelevant to our 1-on-1 Mahjong setting (only one round per game in our test setting). The oracle guiding technique decays the invisible feature during training, which is of independent interest in dealing with imperfect-information. The run-time policy adaptation technique adapts the trained policy at test time, and this may be combined with ACH, which is a training algorithm. In summary, Suphx is more of a novel system than a new algorithm. For this reason, we did not implement and compare Suphx with ACH in our 1-on-1 Mahjong experiment. Nonetheless, the oracle guiding technique may be complementary to ACH in handling imperfect-information, and we will investigate this in future work.

