# OpenReview forum: "Actor-Critic Policy Optimization in a Large-Scale Imperfect-Information Game"
_ICLR.cc/2022/Conference — ICLR 2022 Poster_

### Official Review · Reviewer_m118 · 2021-10-26

**Correctness:** 4
**Technical Novelty And Significance:** 3
**Empirical Novelty And Significance:** 3
**Recommendation:** 8
**Confidence:** 5

**Main Review:**

The paper has a number of strong things to recommend it. The Mahjong results are impressive.  While there a couple of questions below regarding the experiments, in general the experimental design was thorough and reasonable metrics were used. There are, however, a few issues that stand out.

A: "In Theorem 1, we proved that the exploitability of ACH is bounded by when y(a|s; θ t ) is sufficiently close to R t a (s, a)"
Theorem 1 shows a bound on the average policy, which ACH says nothing about tracking. Adding some mechanism for computing an average policy is a non-trivial portion of many of the algorithms in the related work section, and the reason that they could possibly be considered "memory intensive" as noted. Without that, the average online regret experienced in selfplay by ACH might be sublinear, but the policy has no guarantee on exploitability.

B: Algorithm 1 (or other text in the main body) does not include the additional entropy term used in the evaluation. As discussed in the cited work of Hennes et al. (and the work by Perolat et al. that it cites) this is a non-trivial detail in the context of selfplay convergence, and should be directly included in the discussion and pseudocode.

C: There looks to be a mismatch in the experimental results for Kuhn poker and Leduc poker between NeuRD in this paper and in the Hennes et al. paper. Pulling results off of Figure 3 in this paper and Figure 6 of Hennes et al, for Kuhn poker it looks like ~0.2 at 1.5*10^6 iterations vs ~0.03 at 10^5 iterations. There is a smaller mismatch but still non-trivial difference in Leduc poker, with ~0.6 at 2*10^6 iterations vs ~0.4 at 3*10^5 iterations.
Some differences are not necessarily unreasonable, given differences in batch size and other parameters: all algorithms tested in this paper might be expected to improve with more time spent tuning parameters. However, a non-trivial mis-tuning makes it hard to accept the claim that "Clearly, ACH converges significantly faster and achieves a lower exploitability".


There are a number of other smaller issues and questions.

"solve a large-scale two-player zero-sum imperfect-information game"
Mahjong is not solved. This claim in the abstract and conclusion needs to be re-worked.

"An optimal solution to an IIG usually refers to a Nash Equilibrium (NE)"
The scope of this statement should be narrowed, maybe something like  "... optimal solution to a competitive IIG ...". A NE in a cooperative game is usually uninteresting and can be arbitrarily bad with respect to shared utility. For example, a babbling equilibrium in communication games.

"information set size much larger than poker"
The difference in size is mentioned a number times, including both the abstract and conclusion. This would seem to indicate the authors see it as an important quantity, but there is no corresponding detailed discussion of the effects. Why should the reader care about the size of an information set, rather than the number of information sets (corresponding to RL agent states) or possible histories (RL environment states)?

"We are interested in methods that are model-free, i.e., using only trajectory samples to learn, as it is often infeasible to sample multiple if not all actions for each state in large-scale IIGs."
The sentence should be modified so it is clear what state means here. State = agent state / information set? State = environment state / history / game state?
Also consider clarifying the motivation here. Why is the number of actions being sampled tied tied to being model-free? A model-free method certainly advances through an episode with a single action, although it might consider all actions. A model-based method can also operate on one or more sampled trajectories through a game, using the game knowledge as it considers all actions (e.g., Monte Carlo CFR methods). One suggestion would be eliminating the justification given in the text: it is reasonable to just be interested in methods that are model-free.

"Also, we employ deep neural networks to generalize across states without any abstraction, since the state or action abstraction in 1v1 Mahjong is not as straightforward as in Poker, which is explained in the Appendix."
As written, the statement implies that the authors would use state or action abstraction if they were able to do so. Is that true? If so, why? Function approximation would seem to be a more general mapping than a state aggregation. If that's not what the authors intend, consider rewriting to clarify.

"for approaching a NE in 2-player large-scale IIGs"
"... 2-player zero-sum ..."

"We build a 1v1 Mahjong bot, named JueJong, based on a novel neural architecture design and ACH"
Consider dropping "a novel neural architecture design". At some level, every NN architecture proposed for every instance of every problem is novel. If it's not significant enough to spend subtantial time discussing in the body of the text, it's not siginificant enough to make a claim of a novel architecture.

"The reason we use Hedge instead of RM is that softmaxing is shift-invariant, which may be more robust to the function approximation error compared to the threshold operation in RM."
Conversely, RM is scale-invariant, and doesn't require parameter tuning the softmax temperature, which can have a large effect on the performance of Hedge. Was the choice of Hedge vs RM tested? If so, that would be useful information. If not, "may be" seems like a weak justification to switch away from common practice -- are there also other reasons, like ease of implementation in NN frameworks, or ease of theoretical arguments?

How is A^π_t(s,a) being computed? Q(s,a;ω) - π_t(s,.) Q^π_t(s,.;ω)? It would be helpful to state this explicitly, immediately before or after Equation 2 (or in it).

L_π is a computationally intractable sum across all states and actions. Algorithm 1 shows an update sampling s from all states. How is s sampled here?

"Also, M trajectories are sampled according to µ k = (µ_p,k , π_−p,k) at each iteration to estimate E[Ã^π_k (s, a)]."
Is this a typo, or leftover text? k is no longer bound given that the prior paragraph is considering iteration t, and this seems to be the same definition as µ t. Or is this a third set of sampled trajectories used for estimating L_π? If so, don't we need a uniform sample across S given L_π is an unweighted sum across states and actions?

"As shown in Corollary 1, when the behavioral policy is time-invariant, i.e., w_h(s) = f_p^µ_t(s) = w_l(s), ∀s ∈ S, t > 0, the second term in  disappears, and CFR with Hedge is recovered."
The situations where the online learning policy for both learning players is stationary would seem to be limited in scope. Can the authors provide a non-degenerate example where this occurs?

"All the agents we evaluated below use the current policy instead of the average policy, as done in Srinivasan et al. (2018) and Hennes et al. (2020)."
Related to issue B. Suggest re-wording this to something like "the current policy with an entropy term to encourage convergence". As is, the statement is technically correct, but feels like a mischaracterisation in spirit. Both papers, and especially Hennes et al., consider the dynamics of non-stationary policies, and are explicitly modifying the updates to use a single policy rather than just ignoring an averaging step. Most of the experiments in Hennes et al. -- figures 2 to 5? -- seem to be using an average policy.

"We train the best response using PPO with the same hyper-parameters that were used to train the PPO agent."
The static environment for the approximate best response might be qualitatively different than the non-stationary self-play environment, and require very different parameter choices. Can the authors provide the reader with any indication that the parameters used in all of the self play runs are also good choices for PPO-as-best-responder?

"According to the average scores each agent loses to its best response in Figure 1(a), we can conclude that ACH is significantly less exploitable than other methods in the large-scale 1v1 Mahjong environment."
This statement needs to be weakened slightly, given the approximate exploitability results are lower bounds on exploitability: we can certainly conclude that ACH is significantly harder to exploit, but not that it is less exploitable.

It is reasonable to skip comparisons against Suphx, given the slightly different domain, and a possibly non-general algorithm (not clear how it would be applied to Liars dice or the other small games?) However, as a method for generating a strong Mahjong agent, it seems worth mentioning Suphx within the related work section, rather than appearing in the final appendix.


"Poker"
There are many variants of poker. Most attention in the literature has been on Texas hold'em poker. Also, the name of classic games like chess or poker are usually not capitalised, with a few exceptions like Go.

"in the Appendix"
"in Appendix C", "in Appendix D", ...  Help the reader out so they don't need to search the whole appendix.  Occurs multiple times in the paper.

**Summary Of The Paper:**

This paper presents ACH, a neural network policy-gradient method for approximating a Nash equilibrium in two-player zero-sum games. ACH is used to compute a strong agent for a two player variant of Mahjong, competitive with the strongest human players. It has a thorough experimental analysis, both in Mahjong and in smaller test environments.


**Summary Of The Review:**

Taking the core of this paper to be a general policy-gradient update rule for competitive imperfect information environments, with a demonstration of performance in a large game of interest to human players, there is the base of a strong submission. However, the additional theoretical analysis and empirical analysis of small games have a number of issues that do need to be corrected.


------
Post-discussion edit.
The authors have made numerous changes, and I have no remaining concerns with after the Nov 21 revision.

---

> ### Author Response · Authors · 2021-11-19
> **A Response to Reviewer 4**
>
> * Reviewer Comment： The issue with the average policy in ACH and the entropy term
>
> please refer to the summary response at the top.
>
> * Reviewer Comment：The mismatch between the NeuRD results reported in this paper and those published.
>
> please refer to the summary response at the top.
>
> * Reviewer Comment：Mahjong is not solved.
>
> We have changed to use "tackle"
>
> * Reviewer Comment："An optimal solution to an IIG usually refers to a Nash Equilibrium (NE)" The scope of this statement should be narrowed
>
> The scope has been narrowed in the revised paper.
>
> * Reviewer Comment：Why should the reader care about the size of an information set?
>
> Given that the number of histories = the infoset size * the infoset number, we have made this part clearer in the revised paper.
>
> * Reviewer Comment：The sentence about "state" should be modified so it is clear what state means here.
>
> State is used interchangeably with infoset throughout the paper. We have made this part clearer in the revised paper.
>
> * Reviewer Comment：One suggestion would be eliminating the justification given in the text: it is reasonable to just be interested in methods that are model-free.
>
>  We have made this part clearer in the revised paper.
>
> * Reviewer Comment：  As written, the statement implies that the authors would use state or action abstraction if they were able to do so. Is that true? If so, why?
>
> Yes, that is true. The reasons are three-fold. First, our primary goal in this paper is to develop a strong Mahjong AI. Second, both HUNLH poker and 1-on-1 Mahjong are zero-sum IIGs. Third, nearly all successful milestone poker AI (DeepStack, Libratus, etc) use action/state aggregation.
>
> * Reviewer Comment： Consider dropping "a novel neural architecture design"
>
> We have dropped it in the revised paper.
>
>
> * Reviewer Comment： Was the choice of Hedge vs RM tested?
>
> No, it has not been tested. We don't have **strong** justification to use Hedge over RM (will be tested in the future).
>
>
> * Reviewer Comment：How is A^π_t(s,a) being computed? Q(s,a;ω) - π_t(s,.) Q^π_t(s,.;ω)?
>
> Yes, the computation of $A^{\pi_t}(s,a)$ has been added in the revised paper.
>
>
> * Reviewer Comment：Algorithm 1 shows an update sampling s from all states. How is s sampled here?
>
> In practice, we did not train on non-sampled states. This has been made clearer in Section 5 in the revised paper.
>
> * Reviewer Comment："Also, M trajectories are sampled according to µ k ...." Is this a typo, or leftover text?
>
> This is a leftover text.
>
>
> * Reviewer Comment：The situations where the online learning policy for both learning players is stationary would seem to be limited in scope.
>
> Yes, the current policy $\pi_{p,t}$ is rarely stationary.  But for $\mu_{p,t}$, we can set it to the current policy $\pi_{p,t}$ or a stationary sampling policy for which Corollary 1 states a tighter bound on the exploitability. Please check section 5 in the revised paper for more details.
>
>
> * Reviewer Comment："All the agents we evaluated below use the current policy instead of the average policy, as done in Srinivasan et al. (2018) and Hennes et al. (2020)."  Suggest re-wording this to something like "the current policy with an entropy term to encourage convergence".
>
> We have revised the paper according to the reviewer's suggestion
>
> * Reviewer Comment：Most of the experiments in Hennes et al. -- figures 2 to 5? -- seem to be using an average policy.
>
> Yes, NeuRD (Hennes 2020) presented average strategy results in Figs 3, 4, 5, but with either the simple Rock-Paper-Scissors game or the tabular, all actions, counterfactual value NeuRD, **NOT** the NeuRD with a neural network.
>
> * Reviewer Comment：Can the authors provide the reader with any indication that the parameters used in all of the self play runs are also good choices for PPO-as-best-responder?
>
> The static environment for the approximate best response might be qualitatively different than the non-stationary self-play environment. We have made this part clearer in the paper by pointing out the potential risk that parameters used in the self-play PPO might not work satisfactorily with the PPO as-best-responder.
>
>
> * Reviewer Comment："According to the average scores each agent loses to its best response in Figure 1(a), we can conclude that ACH is significantly less exploitable than other methods in the large-scale 1v1 Mahjong environment." This statement needs to be weakened slightly,
>
> We have revised the paper according to the reviewer's comments.
>
> * Reviewer Comment： However, as a method for generating a strong Mahjong agent, it seems worth mentioning Suphx.
>
> Suphx has been mentioned in the Introduction in the revised paper.
>
> * Reviewer Comment： "Poker" There are many variants of poker.
>
> We have changed to be more specific about ‘poker’ in the revised paper.
>
> * Reviewer Comment： "in the Appendix" "in Appendix C", "in Appendix D", ... Help the reader out so they don't need to search the whole appendix.
>
> We have made this clearer in the revised paper.

---

> > ### Comment · Reviewer_m118 · 2021-11-19
> > **after author response**
> >
> > The authors have made significant updates in response to all of the review comments, and the revised paper does a good job addressing a majority of the individual points raised. The response also touches on my three original concerns (A to C), but I think they remain incompletely addressed in the paper. I have expanded my original concerns in this response.
> >
> >
> > Concerns A and B could be grouped together and rephrased as a concern that the algorithm ACH that is presented and analysed is too far from the algorithm that is implemented. This is "only" an issue of writing, but is a non-trivial one, at least in my opinion.
> >
> > I still disagree with the claim in the abstract "We prove the convergence of ACH to a NE under certain conditions", which is restated throughout the paper. Proving ACH & averaging lead to NE is of motivational interest, but ACH as written does not do averaging, and averaging is hard. One can't simply claim to have a solve Z by introducing algorithm X, if it also requires step Y which is impractical. CFR or other iterate-averaging algorithms are not just the iterate generation step, with averaging added on top as a helpful option. The complete algorithms explicitly include that averaging.
> >
> > The gap between theory and practice is widened by putting aside sampling and regularization as empirical choices. The practical implementation introduces bias by not updating on non-sampled states. The regularization on top of the current policy is listed as a choice made for practicality and efficiency.
> >
> > Trying to come up with a suggestion that involves minimum effort, for maximum change. Maybe commit fully to the distinction between the motivation and practical. Start, as is, with a simplified algorithm (NOT called ACH) which includes iterate averaging. The theoretical analysis would then apply to this, but it is clearly not practical. Inspired by this, or modifying it, introduce ACH as the algorithm is now implemented in practice, including regularization and the choice to not update non-sampled states.
> >
> >
> > Regarding the NeuRD results, there was certainly no implication intended that the authors intentionally degraded those results. There are a number of possible reasons for the mismatch: sensitivity to hyperpamater choices as pointed out in the response, errors in the original NeuRD paper, bugs in reimplementation, unlucky samples in a noisy process, or others. However, whatever the reason is, that difference remains and would notably improve NeuRD as a baseline comparison. That uncertainty makes those results less impactful, and moving them to the appendix in favor of the new results is a nice choice.
> > However, that discrepancy from published results should still be noted in appendix F.
> >
> >
> > Finally, a small detail which could still be addressed -- but is NOT a substantial issue.
> > Regarding information set size, I feel like this question is still unanswered. Why should the reader care about the size of an information set, rather than the number of information sets (corresponding to RL agent states) or possible histories (RL environment states)?
> > The closest thing to an answer in the paper is a statement "our benchmark has a larger infoset size (resulting in a larger history size) and a longer game length, compared with existing poker benchmarks". Should "larger history size" be something like "larger number of possible game histories"? Also, Figure 4 would suggest that heads-up no-limit hold'em has more possible histories (around 10^162 * 10^3?) than 1v1 Mahjong (around 10^74 * 10^11). Repeating the question again, why would a reader then care that 1v1 Mahjong has fewer information sets and fewer possible histories than no-limit hold'em, but Mahjong does have a larger information set size? Affects asymptotic complexity of algorithms X, Y, Z in  some particular way? Rules something out?

---

> > > ### Author Response · Authors · 2021-11-20
> > > **A second response to Reviewer 4**
> > >
> > > * Concerns A and B could be grouped together and rephrased as a concern that the algorithm ACH that is presented and analysed is too far from the algorithm that is implemented. This is **only** an issue of writing, but is a non-trivial one, at least in my opinion. Trying to come up with a suggestion that involves minimum effort, for maximum change. Maybe commit fully to the distinction between the motivation and practical. Start, as is, with a simplified algorithm (NOT called ACH) which includes iterate averaging. The theoretical analysis would then apply to this, but it is clearly not practical. Inspired by this, or modifying it, introduce ACH as the algorithm is now implemented in practice, including regularization and the choice to not update non-sampled states.
> > >
> > > We absolutely agree with the reviewer on **this writing issue**, as we have been working on it ever since the beginning of the rebuttal, though still imperfect in the reviewer’s opinion. Based on the reviewer’s suggestion (committing fully to the distinction between the motivation and practical considerations of ACH), we are now working on a new revision that presents a theoretically-motivated algorithm (NOT called ACH) first, then analyses the practical issues when applying it to large-scale problems with deep neural networks, and finally introduces a practical implementation (called ACH) for 1-on-1 Mahjong. The new revision will be uploaded ASAP.
> > >
> > >
> > >
> > > * However, that discrepancy from published results (NeuRD) should still be noted in appendix F.
> > >
> > > The discrepancy has been noted in a new revision, which will be uploaded later.
> > >
> > >
> > >
> > > * Regarding information set size, I feel like this question is still unanswered.
> > >
> > > Initially, we did not think much about the effect of the information set size but just stated a property of 1-on-1 Mahjong compared to poker. Many thanks to the reviewer for encouraging us to delve deeper about the potential influence of the information set size on related algorithms:
> > >
> > > The information set size does not seem to have a direct connection to the convergence of a tabular CFR.However, when trajectory sampling and function approximation are used together, the situation may be different. To be more specific,  in a trajectory sampling algorithm, the variance of the sampled counterfactual values (regrets) of a larger information set may tend to be higher, which may have a large influence on performance when neural network function approximation is used. In other words, 1-on-1 Mahjong may be complementary to poker in evaluating algorithms using deep neural networks and only trajectory samples (the game length in 1-on-1 Mahjong is larger as well).
> > >
> > >
> > > This has been updated in a new revision, which will be uploaded later.

---

> > > > ### Author Response · Authors · 2021-11-21
> > > > **A revision addressing the "writing issue"**
> > > >
> > > > Many thanks again for the reviewer's invaluable suggestion with regard to the presentation of our paper.
> > > >
> > > > We have made a writing revision (the latest uploaded one) to the paper that
> > > >
> > > > * Firstly present a theoretically sound algorithm, called Neural-based Weighted CFR (NW-CFR), with motivations explained (training on sampled advantages instead of sampled counterfactual regrets as done in previous methods) and proofs.
> > > > * Secondly discuss some practical implementation issues of NW-CFR: the average policy and training on non-sampled states.
> > > > * Thirdly present ACH (the initial algorithm) as a practical implementation of NW-CFR.
> > > > * The experiments are based on ACH, demonstrated on 1-on-1 Mahjong, FHP, and small benchmarks from OpenSpiel. The comparing algorithms are Deep CFR, DREAM, PPO, RPG, and NeuRD.
> > > >
> > > > Many thanks again to the excellent work of the reviewer!

---

> > > > > ### Comment · Reviewer_m118 · 2021-11-21
> > > > > **Response to Nov 21 revision**
> > > > >
> > > > > The changes across the couple of revision address all the issues I had concern about. Thank you for taking the comments as suggestions for improvements to the paper.

---

### Official Review · Reviewer_Mmjs · 2021-11-01

**Correctness:** 4
**Technical Novelty And Significance:** 2
**Empirical Novelty And Significance:** 2
**Recommendation:** 6
**Confidence:** 5

**Main Review:**

Actor-Critic Policy Optimization in a Large-Scale Imperfect-Information Game


Positive:

I like that the paper uses 1v1 Mahjong, sounds like an interesting game that is not commonly used in the community. It is also great that the authors approximate exploitability in the large game using a RL method. Furthermore, the prior work is relatively well cited.


Minor Issues:

1) You are missing DeepStack in your introduction, where you list non-tabular methods for large imperfect information games.

2) Corollary 1. Has little very unrealistic assumptions. The current policy is rarely stationary (if it was, often a simple policy gradient method would converge).

Main Issues:


1) My biggest issue with this paper is the proper comparison to prior work  (both in terms of experiments and comparing the methods).


DeepCFR, Double Neural CFR are actually very similar algorithms. Authors state that “All the above methods are model-based in the sense that multiple if not all actions are sampled in a state. For this reason, we do not include these methods for comparison with ACH, which is model-free and uses only trajectory samples to learn.”. This is not true - that line of work builds on top of MCCFR and thus also allows for trajectory samples. The multi-action sampling was just a particular choice for the experiments in DeepCFR paper.

Consider the following paragraph from the DeepCFR paper: “While almost any sampling scheme is acceptable so long as the samples are weighed properly, external sampling has the convenient property that it achieves both of our desired goals by assigning all samples in an iteration equal weight. Additionally, exploring all of a traverser’s actions helps reduce variance. However, external sampling may be impractical in games with extremely large branching factors, so a different sampling scheme, such as outcome sampling (Lanctot et al., 2009), may be desired in those cases.“


2) While the presented theorems are for average policy, the presented algorithm and experiments only use average policy.

This makes the comparison even more unfair, as the prior methods were designed (and reported) using the average policy. Policy averaging in non-tabular settings is a necessary part of a non-tabular algorithm


3) The reported baselines do not match the original publications, and the overall exploitability numbers are quite poor. Consider Leduc, where author’s implementation for both RPG and NeurRD result in exploitability 0.5. Looking at the corresponding publications, these methods are reported to converge to about 0.1.

The authors suggest that “This may due to the neural network approximation error and the fact that we use the current strategy instead of the average strategy for the evaluation.”

First, it is then unfair to report this as exploitability of those methods, as these methods did include a way to average the policy. Current policy has little meaning in these settings. Second, in the case of NeuRD - even the current policy should converge.



**Summary Of The Paper:**

The authors propose a new, non-tabular method for large, two player zero-sum games. The method is evaluated on some standard benchmark games (e.g. Leduc poker) as well as on large game of 1v1 Mahjong.

**Summary Of The Review:**


Summary & Suggestions

To summarize, I think the paper lacks proper comparison to prior methods and makes unsubstantiated claims about  “outperforms related state-of-the-art methods”. Not only are relevant related methods missing, but the ones included do not have the performance previously reported. The authors need to rework the experiments, and also properly compare prior work.

---

> ### Author Response · Authors · 2021-11-19
> **A Response to Reviewer 3**
>
>
> * Reviewer Comment：You are missing DeepStack in your introduction
>
> The DeepStack has been well cited in the revised paper.
>
> * Reviewer Comment：Corollary 1. Has little very unrealistic assumptions. The current policy is rarely stationary
>
> It is indeed that the current policy is rarely stationary.
> Corollary 1 says if the behavior policy $\mu_{p,t}$ (not the current policy $\pi_{p,t}$) is stationary, a tighter bound on the exploitability of the average policy of ACH could be achieved. We have added new experiments with regard to the behavior policy $\mu_{p,t}$ in the revised paper (Appendix E). A preliminary observation is that the performance of the current policy of ACH (trained with an entropy regularization) is not sensitive to the choice of the behavior policy $\mu_{p,t}$ (we can set $\mu_{p,t}$ to the current policy $\pi_{p,t}$ or simply a uniform policy)
>
> * Reviewer Comment：My biggest issue with this paper is the proper
> comparison to prior work (both in terms of experiments and comparing the methods).
>
> New experiments have been added in the revised paper by comparing
> with Deep CFR and DREAM on the FHP benchmark. **The new results show that ACH performs competitively with Deep CFR (using only trajectory samples) and DREAM on the FHP benchmark but consumes significantly fewer samples (please check section 7.3 in the revised paper).**
>
> * Reviewer Comment：The multi-action sampling was just a particular
> choice for the experiments in Deep CFR paper.
>
> Yes, we agree with the reviewer that Deep CFR can run with only trajectory samples.  We have updated the descriptions with regard to Deep CFR and Double Neural CFR in the revised paper.
>
> * Reviewer Comment：This makes the comparison even more unfair,
> as the prior methods were designed (and reported) using the average policy.
>
> We thank the reviewer's comment but believe that the comparison with other methods in our initial submission is **fair**. The method we compared ACH with are NeuRD and RPG, both of which reported experimental results regarding **only** the current policy.
>
> Considering the paragraph from the RPG paper: "Please note that we plot the NASHCONV for the average policy in the case of NFSP, and the current policy in the case of the policy gradient algorithms." and "However, computing the average policy is complex
> and potentially worse with function approximation, requiring storing past data in large buffers."
>
> Considering the paragraph from the NeuRD paper: "Due to the complexity of maintaining a time-average neural network policy to ensure no-regret, we use entropy regularization to induce realtime policy convergence towards the Nash (e.g., as done by Srinivasan et al. , i.e. the RPG paper)."
>
> We did not compare ACH to deep CFR in the initial submission. As replied above, new experimental results on the FHP benchmark have been added in the revised paper, evaluating the average policy in deep CFR but the current policy in ACH, NeuRD, and RPG.
>
> **We sincerely hope the reviewer to reconsider this part of comments.**
>
>
>
> * Reviewer Comment：The reported baselines do not match the original
> publications, and the overall exploitability numbers are quite poor. Consider Leduc, where author’s implementation for both RPG and NeurRD result in exploitability 0.5. Looking at the corresponding publications, these methods are reported to converge to about 0.1.
>
> **For RPG**: the only result about leduc in the RPG paper is the middle-left subfig in Figure 2 [link](https://arxiv.org/pdf/1810.09026.pdf), the final/best exploitability of RPG is around 0.4~0.5. This is further demonstrated in the Openspiel paper [link](https://arxiv.org/pdf/1908.09453.pdf) in its Figure 3, where the final/best exploitability of RPG is around 0.4. Note that in Figure 3 in the Openspiel paper, each line is an average over the **TOP** five seeds. Yet, we report the results of RPG across 8 independent runs with 8 randomly-chosen random seeds. Besides, we use the default implementations together with the default hyper-parameters of RPG in Openspiel for our experiments in Openspiel. **We did not re-implement RPG in the paper.**
>
> **For NeuRD**: please refer to the summary response at the top.
>
> To summarize, we did not see any significant mismatch or contradiction between the results we reported about RPG/NeuRD in our paper and the results in the original RPG/NeuRD papers.
>
> **We sincerely hope the reviewer to reconsider this part of comments**.
>
> * Reviewer Comment：Second, in the case of NeuRD -even the current policy should converge.
>
> After double-checking the NeuRD paper several times
> (examining carefully the Statement 1,Corollary3.1, Corollay3.2, Theorem1, and Theorem2), there is **no** theoretical proof that either the current policy or the average policy of NeuRD itself, **not** CFR(NeuRD), will converge to a NE in **extensive-form** games.

---

> > ### Comment · Reviewer_Mmjs · 2021-11-29
> > **A Response to Response to Reviewer 3**
> >
> > I thank the authors for an extensive response and hard work.
> > I also appreciate that the authors addressed all my comments, and clearly spent non-trivial time adding more experiments.
> > One of my biggest concerns indeed was a lack of baselines that the authors cited, but did not compare against.
> > The authors also agreed that the reasons for not including them were not justified.
> > Furthermore, I agree with the counter-points raised to my last two comments.
> >
> > I also enjoyed the commets of other reviewers, as I share many of them.
> >
> > Overall, I think the authors have greatly improved the paper, so I am raising my score.
> > My worry is that the changes were quite substantial, but I am fine with that.
> >
> > Re NeuRD performance:
> > > "Hence, a reasonable explanation of all the small mismatches may be the NeuRD’s sensitivity to running environments (including hyper-parameters)."
> >
> >  It is completely reasonable explanation, even more so if  you observe such behavior in your implementation. My concern is that the original publication does report a lot lower exploitability, thus the phrase "Experimental results on the proposed 1-on-1 Mahjong benchmark and benchmarks from the literature demonstrate that ACH outperforms related state-of-the-art methods." is maybe too bold?

---

> > > ### Author Response · Authors · 2021-11-29
> > > **Many thanks for the reply.**
> > >
> > > Many thanks for the reply. We really appreciate it !
> > >
> > > We would like to express our sincere gratitude to all the reviewers for their thorough and insightful comments, based on which the paper has been greatly improved since its first submission.
> > >
> > > * reviewer comment: My concern is that the original publication does report a lot lower exploitability, thus the phrase "Experimental results on the proposed 1-on-1 Mahjong benchmark and benchmarks from the literature demonstrate that ACH outperforms related state-of-the-art methods." is maybe too bold?
> > >
> > > Since the conclusion is now mainly based on the experiments in 1-on-1 Mahjong and FHP, the validation of the superior performance of ACH will be made more precise. Also, the small discrepancy between our reported NeuRD results and those in the NeuRD paper has been noted in Figure 11 in the Appendix F.
> > >
> > >
> > > Thanks again for the reviewer's suggestion. We enjoyed the discussion with all the reviewers as well.

---

### Official Review · Reviewer_SULh · 2021-11-02

**Correctness:** 4
**Technical Novelty And Significance:** 3
**Empirical Novelty And Significance:** 4
**Recommendation:** 8
**Confidence:** 4

**Main Review:**

I found the paper to be clear, well organized, and easy to read. The proposed technique, ACH, looks like a good addition to the CFR family of algorithms, separated from Deep CFR in that it can run model-free from trajectories, instead of model-based and requiring subgames. The experiments against human players, and a human champion, in Mahjong is also a nice contribution.

My main concerns are in the evaluation: the lack of a nontrivial poker game to enable a comparison against much of the existing literature, the reliance on Approximate Best Response in Mahjong and treating it as an approximation of exploitability and not a lower bound on exploitability, and a lack of clarity surrounding the average policy in ACH.

The last point is particularly important: with very few exceptions, CFR algorithms such as ACH prove that the average strategy converges, and the current strategy largely does not. Theorem 1 proves that the average strategy converges, but the empirical results only measure the exploitability of the current strategy, and from the results presented the current strategy is very exploitable and thus not a good approximation of a Nash Equilibrium. I’ll expand on this point more below, but the presentation in the paper seems incomplete: the theoretical and empirical results are not reinforcing each other. The paper needs to either describe how to compute the average strategy and present empirical results of both the average and current strategies, or to describe why this is difficult (and a possible weakness in ACH) to justify why no average strategy results are presented.


I’ll now expand on some broader points that highlight what I feel are weaknesses or missed opportunities in the paper and suggest changes, and then list some minor issues probably only of interest to the authors.

Broader comments:

 - Average Strategies. Like I described above, the paper proves the convergence of the ACH average strategy, but only the ACH current strategy is presented in the empirical results. Why? In Sec 5, the authors note that this was also done in Srinivasan 2018 and Hennes 2020, and while I agree about Srinivasan, I do not about Hennes: they do present average strategy results (in their paper, Figs 3, 4, 5). Srinivasan notes that computing the average policy is complex and potentially difficult with function approximation, requiring large buffers to store past strategies.
   - This paper has two weaknesses with respect to the average strategy. First, the empirical results do not show the exploitability of the average strategy, and so (as the authors note in Sec 5.3) we cannot separate ACH’s lack of convergence to 0 exploitability from being due to the current strategy, or due to the neural network approximation error. So, the theoretical contributions and the empirical contributions do not support each other. Second, the body of the text does not explain how the average strategy should be approximated or recovered efficiently, if one wanted to do so. This has been a challenge in related work, such as Deep CFR / Single Deep CFR. If computing or representing the average strategy is indeed a challenge in ACH as it was in Srinivasan 2018, and particularly if the authors had to give up on it and just use the current strategy instead, then the authors need to be more forthright in the text by describing the difficulty and possible solutions, instead of skipping over the issue by saying they follow the evaluation as Srinivasan and Hennes did. And if computing and storing the average strategy is actually easy for ACH, then the empirical results should definitely present it alongside the current strategy results.

 - The paper makes two contributions: presenting ACH as a game-solving algorithm, and presenting 1v1 Mahjong as a testbed game alongside poker (larger information set size, new humans to compete against, etc). However, the paper is less compelling than it could be because it uses Mahjong as the only non-trivial domain for ACH. Heads-Up Texas Hold’em has been the common testbed domain for CFR and related work for almost 15 years. If ACH was demonstrated on a large Hold’em game, ideally the well-studied HULH variant where exact best responses are efficiently computable (thus removing the need for Approximate BR), and then the Mahjong results were presented alongside this standard and large game, then the paper would be a much stronger demonstration of ACH. No-Limit Texas Hold’em would also work, but unfortunately would lose the benefit of exact BR computations. Smaller but still nontrivial poker games are also worth considering: Flop Hold’em, also called [2-round, 4-bet Hold’em], is another benchmark game that just consists of the first two rounds of Heads-up Limit Hold’em. It has 5 card poker hands and an information set size of 1326, making it nontrivial, and yet tabular CFR strategies can be computed on one machine in a day, and exact best responses can be computed in minutes. It would be a compelling testbed for deep RL algorithms like ACH.
   - By focusing the nontrivial-environment empirical analysis of ACH on Mahjong, it unfortunately means that ACH is demonstrated on a game well outside the mainstream in this field, and it’s hard to compare ACH against prior work that has mostly shared a common testbed. The inclusion of Kuhn and Leduc poker are good as a demonstration that the technique is not faulty, but these are tiny, trivial games that can be solved using tabular CFR methods in under a second; Deep Learning approaches like ACH cannot show the benefit of generalization in them, as they could in a larger game like HULH or HUNLH or Flop Hold’em.

   - I am sympathetic to the authors in that the main works they compare to, Srinivasan 2018 and Hennes 2020, also rely on only the tiny games of Kuhn and Leduc as their only poker environments. So I’m not insisting that a larger poker game is a necessary bar to cross for a paper in this area… but to me, it would certainly make the case stronger that ACH works in practice and is worth investigating, if it were demonstrated on a nontrivial poker game that has been used before, and I wish NeuRD and RPG had been similarly demonstrated.


 - The authors have missed some related work: Outcome Sampling CFR, from “Monte Carlo Sampling for Regret Minimization in Extensive Games”, Lanctot et al. Outcome Sampling CFR is a tabular method that learns only from sampled trajectories, as ACH does, instead of requiring full game traversals or sampled subgames as in other CFR variants.


 - When describing Timbers’ Approximate Best Response, the authors should make clear that it doesn’t really approximate an agent’s exploitability. Instead, it provides a *lower bound* on the agent’s exploitability, by creating an opponent that can demonstrably win at least X from the agent. However, the ABR opponent itself suffers from training time, suboptimality from its state representation (abstraction, neural net, etc), and so an exact best response could win more, and in prior work in the poker domain much much more, than an ABR opponent can. Thus, even if Fig 1a had ABR curves converging to 0, it would not be accurate to claim that the strategy was unexploitable and had converged to NE: we would lack evidence of exploitability, but that does not give evidence of lack of exploitability. Using ABR in place of BR makes sense as a fallback in games where BR cannot be tractably computed, but we need to be honest about its limitations. And ideally, the ABR results would themselves be supported by comparisons in smaller nontrivial environments, where an exact BR showed how much value an ABR was missing.


 - Fig 1a. This looks like a convergence graph of the approximate best response: the agent’s policy is fixed, and we’re plotting ABR’s utility against the agent improve over 1e6 steps of training. Is that correct? If so, I would suggest this is not the useful way to show an approximate exploitability graph. What we’re actually interested in is the *agent’s* convergence, similar to 1b, by running a new ABR evaluation every (say) 1e5 training steps, so that we can watch exploitability come down as the agents improve over training, and differentiate if some agents learn faster, slower but reach a lower asymptote, and so on. As you describe, lower exploitability would then be better. The current perspective for Fig 1a, following the ABR over training, may be confusing for the reader who expects the evaluation to focus on the agent.


 - The behavioral strategy, mu_t / mu_k. Throughout the paper, this behavioral strategy used for generating the trajectories was described by its properties (e.g., defn 1). After finishing my review, I tried to find where the paper actually states what mu_t *is* for the strategies computed for the empirical results. Specifically, most CFR algorithms are on-policy (e.g., mu_t is pi_t), and related model-free variants like Outcome Sampling CFR add some epsilon exploration to ensure that all actions are taken with nonzero probability. But Corollary 1 describes mu_p,t as being constant across iterations, which is unlike other CFR variants. So the question of “what is mu_t” is important for understanding the results: did the results use a time-invariant behavioral policy or a policy closer to pi_t? Is the actual mu_t strategy used in the experiments defined in the paper and I just didn’t find it during my post-review skim? Note, for example, that in Algorithm 1 it is not passed as an argument to the function ACH, nor is it described as being computed, like \pi_t is on the 4th line. If mu_t is not described in the main text, then it should be. If it is defined and I have just missed it, then I apologize.

 - In the appendix and conclusion, the authors describe their work as solving 1v1 Mahjong (e.g., “we extend the actor-critic algorithm … to solve a large-scale two-player zero-sum imperfect information game, 1v1 Mahjong”). I understand the authors’ intent in describing this as “solving”, but the word “solve” has a strict technical meaning in game theory and they are only using the word informally. For example, “solving a game” means exactly computing an NE, and “essentially solving” a game means approximating a NE to a very small and exactly computed tolerance (e.g., exploitable for less than 0.001 big blinds/game, in the authors’ cited Bowling 2015 paper). In this paper, neither of these milestones is reached for Mahjong: the ACH policy is not an exact NE (so, not ‘solved’), Fig 1a shows that the ACH policy is still very exploitable by a best response (so, not ‘essentially solved’ as the exploitability value is not small), and even if Fig 1a showed convergence to a value near 0, the “exploitability” calculation itself is by Timbers’ Approximate Best Response, which only gives a lower bound on the true exploitability (i.e., an exact exploitability calculation could still show it to be very exploitable, even if ABR fails to exploit it). So, using “solved” here for Mahjong exaggerates the result, and in an unnecessary way for publication: computing a strong (even if exploitable, or of unknown true exploitability) strategy that is meant to approximate a NE, and that can beat human champions, is already a great achievement.


 - Throughout the paper, the authors frequently use “Poker” when they mean “Heads-up Texas Hold’em”. Poker is a family of related games, including HULH and HUNLH, Kuhn, and Leduc (mentioned in the paper) but also other games like 2-7 Triple Draw, Omaha, Stud, and so on. Statements like “The information set size…[is] 10^3 in Poker” are incorrect. 2-Player Texas Hold’em games do indeed have about 10^3 states in each information set, but Kuhn has 2, Leduc has 5, Heads-up Omaha has 270725, Heads-Up 2-7 Triple Draw has between 2598560 and 9.8*10^12 (depending on how opponent discards are counted), and so on. This is a smaller detail because the authors’ larger point, that 1v1 Mahjong has a larger informations set size *than any poker game* is still true. But other details about “Poker”, like “only two private cards are invisible in Poker”, are only true in the specific game of 1v1 Texas Hold’em Poker.


 - The authors make a distinction between the tabular CFR approach, where a game is abstracted and then the abstract game is solved to a tight tolerance, versus Deep Learning approaches which (as they phrase it) “generalize across states without any abstraction”. The authors should be clear: the neural net is itself a form of abstraction. The hope is that the neural net approach will be more accurate since the feature representation is learned instead of fixed. But it will still be lossy to some degree, as compared to a tabular CFR approach which truly does not do any abstraction, and claims of good generalization have to be demonstrated, not presumed.


 - Table 1. The caption describes 5 independent runs. Does this mean that each agent computation in each row/column was also run 5 times (demonstrating that learning is consistent), or was each agent computed once, and then 5 evaluations of 10k head-to-head games were played? If the latter, and the agents are not trained during evaluation (which I believe is true) then could you instead describe this as 50k independent games, and measure your standard deviation over a larger set? In poker, a technique called ‘duplicate poker’ is commonly used to reduce noise in these head-to-head evaluations, by playing each game twice and switching the agent positions. A similar technique might help for these Mahjong experiments.


Smaller issues:
 - Intro. In the sentence about CFR typically using abstraction techniques, the authors cite Bowling 2015. But that’s an odd citation to use, since that paper describes how abstraction techniques were *not* used, in order to closely approximate a NE. For a description of how abstraction techniques *are* used, maybe try “Evaluating State-Space Abstractions in Extensive-Form Games”, AAMAS 2013, or “A Practical Use of Imperfect Recall”, SARA 2009 from that research group.
 - Intro. ‘current situation of the game’. Suggest ‘current state of the game’.
 - Intro. Suggest replacing ‘bot’ with ‘agent’ in a few spots, to use the standard term.
 - Intro. The authors describe CFR as model-based, but this is not true for all CFR methods; Outcome Sampling CFR (Lanctot et al, NeurIPS 2009) is model-free and, like the authors’ technique, only requires trajectories.
 - Intro. The second-last sentence on Page 1, “abstraction techniques are often manual and require expertise domain knowledge”, has not been true for 10-15 years. The dominant approach to abstraction is simply to use machine learning: namely, using a clustering algorithm like k-means to merge similar states into clusters, based on very simple heuristics that don’t require expert knowledge. See the ‘Evaluating State-Space Abstractions’ reference above, or ‘Potential-Aware Imperfect-Recall Abstraction with Earth Mover’s Distance in Imperfect Information Games’, Ganzfried and Sandholm, AAAI 2014.
 - Sec 2.1. Define ‘infosets’ as ‘information sets (infosets)’ before using the shorthand term.
 - Sec 2.1. This definition of perfect recall appears nonstandard. It defines PR as having the reach probabilities for all h \in I as being the same under policy \pi. The definition I’m familiar with avoids the dependence on any policy: two histories are in an information set if and only if their preceding states also shared the same information sets. This means that an agent never loses the ability to distinguish two histories, thus, perfect recall.
 - Sec 2.1. The authors describe exploitability as the difference between a Nash’s value and a BR’s value. However, a specific useful property of 2-player 0-sum games is that we can compute exploitability without knowing a NE policy, or the game value (the value of a NE in one position against an NE in the other position), because when we sum the BR’s value in each position, we would add and then subtract the same value. So in this setting, it really is just u_p(BR(-p), -p) + u_-p(BR(p), p).
 - Sec 2.1. The total exploitability of a strategy profile defined at the end of 2.1 is not quite what is standard in other papers. The equation given sums the exploitability in each position, but the standard is to then divide by the number of positions (i.e., /2) to return the average loss against a BR across both positions. This is a more natural value because exploitability is then bounded to the range [0, delta] where delta is the maximum loss in the game. For example, if I have 10 chips in poker, my exploitability is then in the range (0, 10). Without dividing by the number of positions, I could lose 10 in one position and 10 in the other position, giving an “exploitability” of 20 when I only have 10 chips. If you do prefer the un-normalized value, related work by Lanctot calls your value “NashConv”, which differs from exploitability by only this division by 2.


**Summary Of The Paper:**

This paper presents Actor-Critic Hedge (ACH): an actor-critic method for approximating Nash equilibrium strategies in large extensive-form games. ACH is an extension of the CFR family of algorithms that uses deep learning, and is able to learn model-free by training on trajectories and not full game traversals or subgames, which is common in much of the related CFR literature. The technique is demonstrated in toy poker domains commonly used in the literature (Kuhn and Leduc poker), Liar’s Dice, and in 1-on-1 Mahjong. In Mahjong, the ACH agent is shown to defeat several human players, including a Mahjong champion.


**Summary Of The Review:**

This is a promising paper: ACH seems like it might be a useful addition to the CFR family of game solving algorithms. My main concerns are in the evaluation. The games used are either trivial domains or, when nontrivial, are not shared by related work, so it’s difficult to compare ACH’s performance. The theoretical results prove the convergence of the ACH average policy (within NN approximation error, etc), but the empirical results only demonstrate the ACH current policy, which has no theoretical convergence guarantees, and does not converge in the experiments shown here. And unless I’ve missed it, the paper does not describe how the average strategy should be approximated by a NN, which has been a challenging topic in related techniques, so the explanation is incomplete. If, like Srinivasan 2018, the authors have decided that computing the average strategy is difficult and so they will only hope that the current strategy (approximately) converges, then they should be more forthright about that shortcoming in this paper.

Overall, I’m landing on ‘marginally below the threshold’, but I’m open to clarification from the authors if I’ve misunderstood elements of the paper. If I have not misunderstood it, then I believe a future draft of this paper that included an empirical analysis of the average strategy, and addressed the smaller issues I noted, could be a strong paper. Including a nontrivial poker domain alongside the Mahjong results would be excellent.

EDIT: Updated review based on Nov 22nd draft
Thank you to the authors for the discussion and for addressing the issues I noted in my review. Specifically:
 - The addition of FHP results, and with exact exploitability figures, is a great addition. The exploitability of roughly 9 chips/game, or 0.09 big blinds/game, is closer to Nash than I had expected.
 - The description of ABR as a lower bound instead of an approximation is clear and consistent.
 - The clarification that 'poker' refers to 'HUNLH' in the page 2 footnote helps, but statements throughout like 'only two private cards are invisible in poker' remain, and could probably be clarified as '...in Texas Hold'em poker' without risking the page limit. Still, it's better than it was.
 - The new section 5, that describes difficulty in using a neural net to represent the average policy and confirms that the presented ACH results use the current policy, is a good clarification.

I've updated several of my scores (correctness 3->4, empirical 3->4, recommendation 5->8). If 7 had been an option I likely would have landed there. The missing piece of the average strategy (a weakness shared by the related work) is still an issue, and it feels like a stretch to describe ACH as 'theoretically justified' (for example, in the abstract) when only the average strategy has theoretical guarantees. But the issue is now acknowledged, and the empirical results in Mahjong and FHP are, to me, compelling enough to recommend acceptance.

---

> ### Author Response · Authors · 2021-11-19
> **A Response to Reviewer 2**
>
> * Reviewer Comment: the lack of a nontrivial poker game to enable a comparison against much of the existing literature
>
> Along with the extensive experimental evaluation around 1-on-1 Mahjong already presented in the paper, we have added new experimental results of ACH with new comparing methods (Deep CFR and DREAM) on a nontrivial poker game you suggested, i.e. FHP in the revised paper. **The new results show that ACH performs competitively with Deep CFR and DREAM on the FHP benchmark yet consumes significantly fewer samples (please check section 7.3 in the revised paper)**.
>
>
> * Reviewer Comment: the reliance on Approximate Best Response (ABR) in Mahjong and treating it as an approximation of exploitability
>
> As the reviewer said, using ABR in place of BR makes sense as a fallback in games where BR cannot be tractably computed. 1-on-1 Mahjong is such a large-scale game (larger than HUNL poker in terms of the number of histories), where BR cannot be tractably computed. **The limitations of ABR that it serves as a lower bound but not an approximation of exploitability have been made clearer in the revised paper.**
>
>
> * Reviewer Comment: a lack of clarity surrounding the average policy in ACH
>
> please refer to the summary response at the top.
>
>
>
>
> * Reviewer Comment: I do not about Hennes: they do present average strategy results (in their paper, Figs 3, 4, 5).
>
>
> Yes, Hennes 2020 presented the average strategy results in Figs 3, 4, 5, but with either the simple Rock-Paper-Scissors game or the tabular, all actions, counterfactual value NeuRD. Hennes 2020 did **not** presented average strategy results with NeuRD with neural networks, considering the paragraph from the NeuRD paper: "Due to the complexity of maintaining a time-average neural network policy to ensure no-regret, we use entropy regularization to induce realtime policy convergence towards the Nash (e.g., as done by Srinivasan et al. )"
>
> Actually, we have also included time-averaged convergence trajectories of different methods on Rock-Paper-Scissors in Figure 8 in the initial submission of the paper.
>
>
>
> * Reviewer Comment: The authors have missed some related work: Outcome Sampling CFR
>
> Thank you for the excellent pointers to related work! we have added it in the revised paper.
>
>
>
> * Reviewer Comment: Fig 1a. This looks like a convergence graph of
> the approximate best response: the agent’s policy is fixed, and we’re plotting ABR’s utility against the agent improve over 1e6 steps of training. Is that correct?
>
>
> Yes, it is correct. We showed the training process of the ABR,
> not the agent's convergence. It is indeed more appropriate to plot the agent's convergence by running a new ABR evaluation every (say) 1e5 training steps, as you suggested. Yet, considering a single ABR evaluation cost, in our 1-on-1 Mahjong setting, is 800 CPUs running about 3-4 days, the more appropriate evaluation is just too expensive for us. Currently, we may have to stay with the ABR evaluation. We have made this clearer in the revised paper.
>
>
> * Reviewer Comment: what is $\mu_t$ for the strategies computed for the empirical results.
>
> The behavior policy $\mu_{p,t}$ can be considered as a hyper-parameter of
> ACH. For $\mu_{p,t}$, we can set it to the current policy $\pi_{p,t}$ or a constant/stationary sampling policy for which Corollary 1 states a tighter bound on the exploitability. We have used $\mu_{p,t} =\pi_{p,t}$ for the experimental results. We have made this clearer in Section 5 in the revised paper.
>
>
> * Reviewer Comment: the use of the word **solve**
>
> We have changed to use "tackle"
>
>
> * Reviewer Comment: different variants of Poker
>
> We have changed to be more specific when we talk about poker games in the revised paper.
>
>
> * Reviewer Comment: it will still be lossy to some degree, as compared
> to a tabular CFR approach which truly does not do any abstraction, and claims of good generalization have to be demonstrated, not presumed
>
> We are aware that neural network is a form of approximation.
> We agree that claims of good generalization have to be demonstrated, not presumed. We have changed to state that ACH generalizes cross states using a neural network in the revised paper.
>
>
>
> * Reviewer Comment: The caption describes 5 independent runs.
> Does this mean that each agent computation in each row/column was also run 5 times (demonstrating that learning is consistent), or was each agent computed once, and then 5 evaluations of 10k head-to-head games were played?
>
> The caption means each agent computation was run 5 times. But, the i-th run of ACH is evaluated against only the i-th run of other methods. So, there are totally 5 independent runs for each entry in Table 1. We have made this clearer in the revised paper.
>
> The 'duplicate' technique was also used in our 1v1 Mahjong evaluation, same as the "duplicate poker" technique.
>
>
> * Reviewer Comment: All the samller issues mentined by the reviewer
>
>
> All the smaller issues have been incorporated in the revised paper.

---

> > ### Comment · Reviewer_SULh · 2021-11-19
> > **Unit for exploitability**
> >
> > Thank you for your response - it sounds like a comprehensive set of edits. I've started reading the revised doc.
> >
> > One quick question. For the new FHP exploitability results in Fig 3, what is the unit in which exploitability is being measured? I didn't see this defined, and in the CFR / computer poker literature there are three reasonable units it could be:
> > - big blinds/game (FHP) or antes/game (Kuhn, Leduc)
> > - Chips/game, where a big blind is (say) 2 chips and a small blind is 1 chip.
> > - milli-blinds/game, particularly common in the CFR poker literature, where 1000 milli-blinds/game equals 1 big blind/game.
> >
> > Always Fold is a trivial strategy in FHP, and is exploitable for 0.75 big blinds/game, or 750 milli-blinds/game. In Fig 3a, all of the tested agents look like they're in the range 7 units to 10 units of exploitability. So depending on the unit (if it's one of these three options), the agents are either very far or very close to Nash.

---

> > > ### Author Response · Authors · 2021-11-20
> > > **Response to "the unit for exploitability"**
> > >
> > > Thanks so much for the quick reply. We really appreciate it.
> > >
> > > For the new FHP exploitability results in Fig 3, **the unit in which exploitability is being measured is chips/game, where a big blind is 100 chips and a small blind is 50 chips.** Always Fold is exploitable for 75 chips/game in our measure.
> > >
> > > We have made changes accordingly, and will update the revision later. Many thanks again for the comment.

---

### Official Review · Reviewer_8wVT · 2021-11-03

**Correctness:** 4
**Technical Novelty And Significance:** 3
**Empirical Novelty And Significance:** 3
**Recommendation:** 5
**Confidence:** 3

**Main Review:**

What I would like to see most in this paper is two things: more tabular analysis and experiments and better comparisons to existing methods.

This method is essentially doing regression on a weighted CFR. However, this weighted CFR is a new method (to my knowledge). What properties does it have compared to normal CFR? Can we run experiments with tabular weighted CFR and compare it to normal CFR? Why might we prefer weighted CFR to normal CFR? Answers to these questions would help answer other questions I have such as why didn’t the authors compare to Deep CFR/DREAM? The authors claim that they are not related because they are not a policy gradient method, but the proposed method isn’t really a policy gradient method either. In fact I find it closer to Deep CFR than to NeuRD/RPG, but I do see the similarities with those methods as well.




**Summary Of The Paper:**

This paper proposes a method somewhat similar to Deep CFR, NeuRD and RPG, where a policy is trained to predict a weighted counterfactual regret. In small openspiel experiments the method seems to outperform NeuRD and RPG, but no comparisons are made to Deep CFR/DREAM/ARMAC. Impressive results against a top Mahjong player demonstrate that this method can scale to large games.


**Summary Of The Review:**

Overall I find the experiments against the top Mahjong player to be very compelling and am excited by this research direction. However, I think this paper skips some crucial steps in analyzing the tabular method that this is based on. If the authors are able to address this issue and explain why we would want to use this method vs. deep CFR/DREAM or compare to those methods then I would consider raising my score.

---

> ### Author Response · Authors · 2021-11-19
> **A Response to Reviewer 1**
>
> * Reviewer Comment: more explanation and analysis of the
> weighted CFR introduced in the paper.
>
> Response: To be more precise, the weighted CFR introduced in this paper is a **family** of CFR algorithms, with each setting of the weight $w_t(s)$ being an instance of a CFR algorithm. For instance, the normal CFR algorithm is a type of weighted CFR with $w_t(s)= 1.0, \forall t, s$.
> The **primary** reason to introduce the weighted CFR in this paper is to provide a theoretical tool to analyse the convergence property of ACH, since ACH in expectation approximates a particular type of weighted CFR with $w_t(s) = f^{\mu_t}_{p}(s)$ (as you said ACH is essentially doing regression on a weighted CFR). We have clarified this part in the revised paper.
>
> * Reviewer Comment: explain why we would want to use this method (our method ACH) vs. deep CFR/DREAM or compare to those methods
>
> Response: The initial reason we did not include Deep CFR/DREAM for comparison is that Deep CFR/DREAM (samples at each iteration must be stored and trained) are more computation-intensive than ACH (using only samples of the current iteration), which may not be as scalable as ACH for our 1-on-1 Mahjong. The above reason may not sound too persuasive, and we agree that the comparison between ACH and Deep CFR/DREAM could significantly improve the paper. Hence, we have now added the corresponding experimental comparison in the revised paper. The new results show that **ACH performs competitively with Deep CFR and DREAM on the FHP benchmark yet consumes significantly fewer samples (please check section 7.3 in the revised paper)**.

---

> > ### Comment · Reviewer_8wVT · 2021-11-22
> > **Tabular W-CFR**
> >
> > Thank you for adding the additional experiments, I think the paper looks much better now and I am almost ready to vote for acceptance. However, I think the paper is still missing analysis on tabular weighted-CFR. Could the authors run some tabular experiments comparing W-CFR to CFR? I think that would help the paper a lot more. Also, I missed that the paper uses the last iterate instead of the average. I think this is a big issue so that makes me hesitant to accept as well.

---

> > > ### Author Response · Authors · 2021-11-23
> > > **Response to the experiments on Tabular Weighted CFR and the last iterate instead of the average policy used&evaluated in ACH.**
> > >
> > > Many thanks for the reply. We really appreciate it!
> > >
> > > * **Reviewer comment**: Could the authors run some tabular experiments comparing W-CFR to CFR?
> > >
> > > Response:  Yes, we can. We are working on this and will upload the corresponding revision ASAP.  As stated, CFR is an instance of Weighted CFR (W-CFR) with the weight $w_t(s)$ set to 1.0. We will present experimental results with different settings of $w_t(s)$ in W-CFR.
> > >
> > > * **Reviewer comment**: I missed that the paper uses the last iterate instead of the average policy in ACH.
> > >
> > > Response:  The reasons we use the last iterate instead of the average policy in ACH are two-fold:
> > >
> > > 1) As pointed out in NeuRD, RPG, and Perolat 2021-ICML (From Poincar´e Recurrence to Convergence in Imperfect Information Games), **obtaining the average policy with deep neural nets in large-scale games is inherently difficult**, either due to the computation or the memory demand. As a result, results were reported with the **last iterate policy** in NeuRD and RPG.
> > >
> > > Considering the paragraph from the NeuRD paper: "**Due to the complexity of maintaining a time-average neural network policy** to ensure no-regret, we use entropy regularization to induce realtime policy convergence towards the Nash (e.g., as done by Srinivasan et al. , i.e. the RPG paper)."
> > >
> > > Considering the paragraph from the RPG paper: "Please note that we plot the NASHCONV for the average policy in the case of NFSP, and the current policy in the case of the policy gradient algorithms." and "However, **computing the average policy is complex and potentially worse with function approximation**, requiring storing past data in large buffers."
> > >
> > > Considering the paragraph from the Perolat 2021-ICML paper: “these convergence guarantees are not conducive to learning in large games, which rely on general function approximation techniques (e.g., deep neural networks) that are **inherently difficult to time-average**.”
> > >
> > >
> > > 2) On the other hand, for large-scale problems, we could give up the idea of computing an average policy but turn to some additional technique to (hopefully) induce the current policy convergence towards a NE. RPG and NeuRD took this direction by adding an entropy regularization to the current policy training, which was, to some extent, theoretically justified in the recent Perolat 2021-ICML paper. We decided to take this direction for ACH as well.
> > >
> > >
> > >
> > > Also, we have made it (the use of last iterate policy in ACH) clearer by re-organizing the paper (the Nov 22 revision, i.e., the last revision):
> > >
> > > (1)Firstly present a theoretically sound algorithm, called Neural-based Weighted CFR (NW-CFR), with motivations explained (training on sampled advantages instead of sampled counterfactual regrets as done in previous methods) and proofs.
> > >
> > > (2)Secondly discuss some practical implementation issues of NW-CFR: **the average policy** and training on non-sampled states.
> > >
> > > (3)Thirdly present ACH (**using the last-iteration policy**) as a practical implementation of NW-CFR.
> > >
> > > (4)The experiments are based on ACH, demonstrated on 1-on-1 Mahjong, FHP, and small benchmarks from OpenSpiel. The comparing algorithms are Deep CFR, DREAM, PPO, RPG, and NeuRD. **Note that results are reported for the last-iterate policy in ACH, PPO, NeuRD, and RPG and the average policy in Deep CFR and DREAM.**
> > >
> > > **Many thanks** again for the invaluable suggestions and hope the above response addresses your concerns with regard to the tabular Weighted CFR and the use of the last iterate policy (instead of the average policy) in ACH.

---

> > > > ### Author Response · Authors · 2021-11-23
> > > > **New experimental results with regard to the Weighted CFR and CFR are added in the Appendix D**
> > > >
> > > > Many thanks again for the invaluable suggestion of the reviewer.
> > > >
> > > > New experimental results with regard to the Tabular Weighted CFR and CFR are added in the **Appendix D** in the latest revision. The exploitability results on Kuhn poker, Leduc poker, and Liar’s Dice validate the theoretical claims of the paper in Theorem 1 and Corollary 1.
> > > >
> > > > Also, we would like to emphasize that the introduction of the weighted CFR (Definition 1) in this paper is to **provide a theoretical tool** to better analyse the theoretical property of the proposed Neural-based Weighted CFR (NW-CFR) and the proposed algorithm ACH (ACH is a practical implementation of NW-CFR).  The superior performance of ACH is demonstrated on 1-on-1 Mahjong, FHP, and three IIGs in OpenSpiel, in comparison with Deep CFR, DREAM, PPO/A2C, RPG, and NeuRD.

---

> > > > > ### Comment · Reviewer_8wVT · 2021-11-23
> > > > > **Tabular Results**
> > > > >
> > > > > Thanks a lot for running these experiments so fast. My worry here is that the purple "current" line doesn't converge. Isn't this current algorithm what NW-CFR is based on? And isn't uniform pretty much normal CFR?

---

> > > > > > ### Author Response · Authors · 2021-11-24
> > > > > > **A Response to the "Tabular Results" from a Board Perspective**
> > > > > >
> > > > > > Many thanks for the quick reply. We really appreciate it!
> > > > > >
> > > > > > Before responding directly to the reviewer’s comments, we would like to reply from a boarder perspective, with regard to the **motivation** of Neural-based Weighted CFR (NW-CFR), the **relationship** among NW-CFR, the tabular Weighted CFR(W-CFR), and ACH(a practical implementation of NW-CFR), and **the fundamental reason** for the superior performance of ACH compared to Deep CFR on FHP.
> > > > > >
> > > > > >
> > > > > > As stated in Section 3 in the paper, NW-CFR is a new neural-based CFR algorithm. The fundamental difference between NW-CFR and other neural-based CFR algorithms (Deep CFR, Dream, Double neural CFR, etc.) is that NW-CFR accumulates and **trains on the sampled advantage $\tilde{A}^{\pi_k}(s, a)$ instead of the sampled counterfactual regret $\tilde{r}^c_k(s, a)$ in other methods**. As explained in the last paragraph in Section 3, the sampled advantage can have a significantly lower variance than that of the sampled counterfactual regret, $\tilde{r}^c_k(s, a)=[f^{\mu_{k}}_{p}(s)]^{-1}\tilde{A}^{\pi_k}(s, a)$, especially in games with long episodes.
> > > > > >
> > > > > > As the reviewer pointed out and we demonstrated in Section 4, NW-CFR is a straightforward neural extension to a type of weighted CFR. **To better analyse the theoretical property of NW-CFR, we defined this W-CFR in definition 1**. Moreover, we provided the theoretical results on W-CFR (equivalently on NW-CFR under certain conditions) for any behavior policy $\mu_{p,t}$ (Theorem 1) and a stationary behavior policy (Corollary 1). **W-CFR is the tabular correspondence of NW-CFR and only serves as a theoretical tool. We are not claiming that W-CFR is better than CFR. CFR is an instance of W-CFR with the weight $w_t(s)=1.0$**.
> > > > > >
> > > > > > As our primary goal is to solve 1-on-1 Mahjong, in which the calculation of the average policy is impractical (explained and justified in Section 5), **we develop a practical implementation of NW-CFR: ACH**. ACH adds an entropy regularization (as RPG and NeuRD did) to hopefully induce the convergence of the current policy $\pi_{p,t}$. **The main purpose of the entropy regularization** is to induce the convergence of the current policy and avoid the calculation of the average policy in large-scale games.
> > > > > > **A side effect** of the entropy regularization is regularizing the current policy towards the uniform policy.
> > > > > >
> > > > > > **ACH performs, in terms of exploitability, better and converges 1000 times faster than Deep CFR on FHP**. This is because, from our point of view, the lower variance of the sampled advantages allows ACH to train on samples on-the-fly. Yet, at each iteration, Deep CFR has to train on samples collected in all previous iterations. ACH is also evaluated extensively on 1-on-1 Mahjong and benchmarks from Openspiel with other methods (PPO, RPG, and NeuRD). In all experiments, ACH demonstrates the best performance.
> > > > > >
> > > > > > The results on FHP demonstrated the superiority of training on the sampled advantages over the sampled counterfactual regrets. All the experimental results validate the approximations ACH makes to NW-CFR in order to deal-with large scale problems with deep neural networks and only trajectory samples.
> > > > > >
> > > > > > **To summarize, the better experimental results of ACH and the potential advantage of NW-CFR over other neural-based CFR algorithms (Deep CFR, DREAM, etc) is not because W-CFR (NW-CFR and ACH are derived from W-CFR) has any advantage over CFR (Deep CFR and DREAM are derived from CFR), but because NW-CFR and ACH train on the sampled advantages instead of the sampled counterfactual regrets used in those methods.**

---

> > > > > > ### Author Response · Authors · 2021-11-24
> > > > > > **A Response to the "Tabular Results" answering specific questions.**
> > > > > >
> > > > > > * Review comment：My worry here is that the purple "current" line doesn't converge. Isn't this current algorithm what NW-CFR is based on? And isn't uniform pretty much normal CFR?
> > > > > >
> > > > > > NW-CFR is a straightforward neural extension to W-CFR, so the behavior policy **$\mu_{p,t}$ can be viewed as a hyper-parameter in NW-CFR.**
> > > > > >
> > > > > > The tabular experiments in the appendix D is to investigate the convergence of the **tabular** W-CFR under different settings of $\mu_{p,t}$. When $\mu_{p,t}$ is uniform, W-CFR performs similarly as CFR, which is what Corollary 1 states. When $\mu_{p,t}$ is $\pi_{p,t}$, which varies across iterations, Theorem 1 says the exploitability may be large, and this is demonstrated in the “Current” lines in Figure 9. However, when $\mu_{p,t}$ is a mixed strategy between $\pi_{p,t}$ and the uniform strategy, the range of $\mu_{p,t}$ is reduced. As Theorem 1 suggests, the exploitability gets lower as the range of $\mu_{p,t}$ across iterations gets tighter. This is experimentally demonstrated in the “Current(x)” lines in Figure 9. This, to some extent, justifies the good performance of ACH with $\mu_{p,t}=\pi_{p,t}$, **because the current policy $\pi_{p,t}$ is regularized towards a uniform strategy by the entropy loss in ACH.**
> > > > > >
> > > > > > Theoretically, it might be better to use the uniform sampling strategy for $\mu_{p,t}$ in ACH, if we can use the average strategy for evaluation. Yet, the average strategy is not available in ACH, and we use the last-iteration policy for evaluation in ACH. As a result, we have conducted experimental studies with regard to the choice of $\mu_{p,t}$ in ACH on FHP in the Appendix F. The results showed that the performance of ACH when $\mu_{p,t}$ is uniform is close with that when $\mu_{p,t}=\pi_{p,t}$. Considering the benefit of training the value and the policy nets using the same trajectory samples when $\mu_{p,t} =\pi_{p,t}$, the behavior policy $\mu_{p,t}$ is set to $\pi_{p,t}$ in ACH for all the experiments.
> > > > > >
> > > > > >
> > > > > > Hope the above addresses the reviewer’s concern, and **we are open for any further clarification and discussion**. Many thanks again for the hard work of the reviewer.

---

> > > > > > ### Author Response · Authors · 2021-11-27
> > > > > > **Many thanks and a gentle reminder that the final stage of discussion closes at 29th.**
> > > > > >
> > > > > > Many thanks again for the instructive comments of the reviewer.
> > > > > >
> > > > > > As the final stage of discussion ends at 29th, we are eagerly to know whether our previous responses have clarified your concerns. If not, we would be happy to discuss this further with you.
> > > > > >
> > > > > > Really sorry for any disturbance caused, and Happy Thanksgiving Day if the reviewer is currently on this holiday.

---

### Author Response · Authors · 2021-11-19
**A summary of the rebuttal to all reviewers**

We thank the reviewers for their insightful and thorough feedback. We are encouraged that all reviewers find our results on **1-on-1 Mahjong (the FOCUS of the paper)** impressive (R1), very compelling (R1), a nice contribution (R2),  interesting (R3), impressive (R4) and a base of a strong submission (R4). Before addressing specific concerns, we would like to emphasize that the bulk of efforts and primary goal of this paper are to develop a strong 1-on-1 Mahjong AI, and the development of the new algorithm ACH serves primarily for this goal. The primary goal is the basis of all the important decisions made:
* Using neural networks to generalize across states instead of abstraction techniques, because abstractions may be much more difficult in 1-on-1 Mahjong than in poker.
* The benchmarks tested.
* The methods compared with should be memory&computation efficient, similar to PPO, NeuRD, and RPG, and most importantly rely only on **trajectory** samples. (for a single 40-step 1-on-1 Mahjong game, sampling 2 actions in a state results in 2^40 samples, which is prohibitive).
* Using the current policy instead of the average policy for evaluation, because computing the average policy with neural networks for large-scale problems is extremely difficult, as described in the RPG paper (Srinivasan 2018-NIPS), the NeuRD paper (Hennes 2020-AAMAS), and Perolat 2021-ICML .


There are **four primary concerns** from the reviewers, all of which are with regard to the experiments. We address them below and have made changes to the paper accordingly.

* **Concern 1**: a nontrivial poker game should be investigated.

Along with the extensive experimental evaluation around 1-on-1 Mahjong already presented in the paper, we have now added new experimental results of ACH with new comparing methods (Deep CFR and DREAM) on a nontrivial poker game, i.e., FHP in the revised paper. The new results show that **ACH performs competitively with Deep CFR and DREAM on the FHP benchmark yet consumes significantly fewer samples** (**please check section 7.3 in the revised paper**).

* **Concern 2**: Other CFR motivated methods (such as Deep CFR and DREAM) should be included.
see above

* **Concern 3**:The mismatch between the NeuRD results we reported and those in the NeuRD paper.

**The code for NeuRD is not publicly available**, considering that the implementation status of NeuRD in Openspiel is "X: known problems; please see github issues." [link](https://github.com/deepmind/open_spiel/blob/master/docs/algorithms.md).

As a result, we have to implement NeuRD by ourselves. The way we implemented NeuRD is adapting the openspiel RPG implementation, by changing the update of the policy along with some other details mentioned in the NeuRD paper.

We are aware the small mismatch in the experimental results for kuhn poker and leduc between our NeuRD results and the results in the NeuRD paper, as pointed out by two reviewers. In the meantime, we also notice a small mismatch of the NeuRD results between the NeuRD paper (Fig 6 (a) (b) (c), [link](https://arxiv.org/pdf/1906.00190v4.pdf) ) and the Openspiel paper (Fig 5, [link](https://arxiv.org/pdf/1908.09453.pdf) ). Hence, a **reasonable** explanation of all the small mismatches may be the NeuRD’s sensitivity to running environments (including hyper-parameters).

We did **NOT** do any mis-tuning to make the results of NeuRD in our paper to look bad. Our NeuRD code is available as a supplementary zip file. The hyper-parameters are listed in Appendix G3.

* **Concern 4**: Theoretical results are provided with the average policy in ACH, yet experimental results are reported only for the current policy in ACH

When it comes to the average policy, we essentially face the same problem as the RPG paper (Srinivasan 2018-NIPS) and the NeuRD paper (Hennes 2020-AAMAS). As pointed out in NeuRD, RPG, and Perolat 2021-ICML (From Poincar´e Recurrence to Convergence in Imperfect Information Games), obtaining the average policy with deep neural nets in large-scale games is inherently difficult.

On the other hand, we could give up the idea of computing an average policy but turn to some additional technique to (hopefully) induce the current policy convergence towards a NE. RPG and NeuRD took this direction by adding an entropy regularization to the current policy training, which was, to some extent, theoretically justified in the recent Perolat 2021-ICML paper. We decided to take this direction for ACH as well.

To summarize, there indeed was a lack of clarity surrounding the average policy in ACH in the initial submission of the paper. We now have updated the paper accordingly (please check section 5 in the revised paper).

* We address reviewers other comments below respectively and have incorporated all feedback. **Many thanks** again for the invaluable hard work of all reviewers to improve our paper.

---

> ### Author Response · Authors · 2021-11-21
> **A writing revision is uploaded**
>
> We have made a writing revision (the latest uploaded one) to the paper that
>
> * Firstly present a theoretically sound algorithm, called Neural-based Weighted CFR (NW-CFR), with motivations explained (training on sampled advantages instead of sampled counterfactual regrets as done in previous methods) and proofs.
> * Secondly discuss some practical implementation issues of NW-CFR: the average policy and training on non-sampled states.
> * Thirdly present ACH (the initial algorithm) as a practical implementation of NW-CFR.
> * The experiments are based on ACH, demonstrated on 1-on-1 Mahjong, FHP, and small benchmarks from OpenSpiel. The comparing algorithms are Deep CFR, DREAM, PPO, RPG, and NeuRD.
>
> Many thanks again to the excellent work of all the reviewers!

---

> > ### Author Response · Authors · 2021-11-23
> > **A new revision is uploaded with the addition of experimental results of the weighted CFR. Changes happen in the Appendix D.**
> >
> > Many thanks again for the excellent work of all reviewers. We really appreciate it.
> >
> > New experimental results with regard to the Tabular Weighted CFR and CFR are added in the **Appendix D** in the latest revision. The exploitability results on Kuhn poker, Leduc poker, and Liar’s Dice validate the theoretical claims of the paper in Theorem 1 and Corollary 1.
> >
> > Also, we would like to emphasize that the introduction of the weighted CFR (Definition 1) in this paper is to **provide a theoretical tool** to better analyse the theoretical property of the proposed Neural-based Weighted CFR (NW-CFR) and the proposed algorithm ACH (ACH is a practical implementation of NW-CFR). **The superior performance of ACH is demonstrated on 1-on-1 Mahjong, FHP, and three IIGs in OpenSpiel, in comparison with Deep CFR, DREAM, PPO/A2C, RPG, and NeuRD.**

---

### Public Comment · ~Zhenjie_Zhao1 · 2023-11-23
**I am having difficulty understanding the relationship between the first term of equation 29 and equation 2**

Could you please provide further clarification or explanation to help me better comprehend their connection?

To clarify the differences between equation 2 and the first term of equation 29, let's compare them directly. In equation 2, there is a square function applied, while the first term of equation 29 does not include a square term. The variable "y" is used to approximate "A," but it may seem unclear why they are multiplied together in the first term of equation 29. Regarding the use of "c" as a clipping operator, it appears that there may be a missing parentheses after it.

Thanks.

---

### Decision · Program_Chairs · 2022-01-20

**Decision:**

Accept (Poster)

**Comment:**

This paper presents a Actor-Critic Hedge (ACH) method for 1-on-1 Mahjong. It is is an actor-critic method for approximating Nash equilibrium strategies in large extensive-form games. ACH extends the CFR family of algorithms that uses deep learning and model-free training (not using full game traversal). The propose ACH agent defeats several human players, including a Mahjong champion. This is impressive.

The reviewers and authors have extensive discussions and the authors managed to address most of the concerns from the reviewers. The overall opinions from the the reviewers favor acceptance. Below are some of the strength and weakness summarized from the reviewers:

Strength:
* Extensive experiments and impressive performance.
* New policy based algorithm for competitive environments.
* Reviewers' questions are well addressed.

Weakness:
Lack of more tabular theoretical analysis. Need experiments to compare to existing methods. Theory and experiment does not match.